

# Assessing Vertical Coordinate System Performance in the Regional Modular Ocean Model 6 configuration for Northwest Pacific

Inseong Chang[1], Young Ho Kim[*1], Young-Gyu Park[2], Hyunkeun Jin[2], Gyundo Pak[2], Andrew C. Ross[3], Robert Hallberg[3]

[1]Division of Earth Environmental System Science, Pukyong National University, Busan, Republic of Korea
[2]Ocean Circulation and Climate Research Department, Korea Institute of Ocean Sciences and Technology, Busan, Republic of Korea
[3]NOAA OAR Geophysical Fluid Dynamics Laboratory, Princeton, NJ, USA

*Correspondence to*: Young Ho Kim (yhokim@pknu.ac.kr)

**Abstract.** The Northwest Pacific (NWP) has a complex ocean circulation system and is among the regions most affected by climate change. To facilitate rapid responses to marine incidents and effectively address climate variability impacts, the Korea Institute of Ocean Science and Technology (KIOST) developed the Korea Operational Oceanographic System–Ocean Predictability Experiment for Marine Environment (KOOS-OPEM), a high-resolution regional ocean prediction system based on Modular Ocean Model version 5 (MOM5). In this study, the base model of KOOS-OPEM was upgraded to MOM6 to enhance its regional ocean modeling capabilities. A key advancement of MOM6 is its flexible vertical coordinate system enabled by a Lagrangian remapping system. Taking advantage of this feature, we evaluated the impact of vertical coordinate choices on model performance by comparing the HYBRID (z*-isopycnal) and ZSTAR (z*) configurations. Model outputs from the 2003-2012 period were assessed against multiple observational datasets and reanalysis products to determine their ability to reproduce key oceanographic features. The results indicated that HYBRID better preserved stratification and reduced spurious diapycnal mixing, significantly improving the representation of North Pacific Intermediate Water (NPIW). In contrast, ZSTAR exhibited excessive diapycnal mixing, resulting in a thicker isopycnal layer associated with NPIW and a salinity bias of approximately 0.2 psu. An idealized age tracer experiment further confirmed that ZSTAR facilitates excessive downward diffusion of younger surface waters, eroding the minimum salinity layer of the NPIW. For sea surface temperature (SST), both configurations captured seasonal patterns but exhibited biases. HYBRID produced a stronger warm bias in the Kuroshio-Oyashio Transition Zone. In the East/Japan Sea, ZSTAR displayed an exaggerated warm bias exceeding 3℃, primarily due to an overshoot of the East Korea Warm Current (EKWC), which extended too far north without proper separation. In tidal simulations, HYBRID outperformed ZSTAR in reproducing M2 tidal amplitudes in the Yellow Sea, where stratification plays a key role. Conversely, ZSTAR underestimated these amplitudes due to its limitations in representing stratification. Despite its advantages, HYBRID underperformed in high-latitude regions, exhibiting larger temperature and salinity biases between 100 m and 600 m depth, with temperature biases reaching approximately -1℃. This discrepancy arose because HYBRID maintained fewer active layers in weakly stratified regions, reducing vertical resolution and leading to errors in water mass representation. To mitigate these issues and improve HYBRID's performance in high-latitude regions, adjustments to the maximum layer thickness and target density profiles are necessary. Beyond physical



ocean modeling, integrating biogeochemical components is essential for advancing the understanding of ecosystem
dynamics in the NWP.

## 1 Introduction

The Northwest Pacific (NWP) Ocean has a complex circulation system characterized by strong western boundary currents,
including the Kuroshio and Oyashio Currents, which exhibit significant energetic variability. This region also encompasses
several marginal seas, including the South China Sea, the East/Japan Sea, the Yellow Sea, and the East China Sea, which are
interconnected through narrow straits (see Fig. 1). Each marginal sea exhibits unique physical oceanographic characteristics
shaped by its complex bottom topography and hydrodynamic processes. For instance, the South China Sea, with its deep
basins and intricate current system, is strongly influenced by seasonal monsoons and the intrusion of the Kuroshio Current.
The East/Japan Sea, a semi-enclosed deep marginal sea with steep underwater topography, shares characteristics with major
oceans, including a regional western boundary current (the East Korea Warm Current, EKWC), an intermediate salinity
minimum layer, deep water formation via its own ventilation system, and both mesoscale and sub-mesoscale eddies and
fronts (Kim and Kim, 1983; Ichiye et al., 1984; Senjyu, 1999; Kim et al., 2001). The East China Sea, characterized by a
broad continental shelf and shallow waters, has circulation patterns driven by wind variations, tides, riverine discharges, and
external forcings from the Taiwan and Tsushima Straits, along with Kuroshio Current intrusions (Isobe, 2008; Gan et al.,
2016). The Yellow Sea, known for its shallow depths and extensive tidal flats, is dominated by strong tidal currents and
seasonal temperature variations, which drive vertical mixing and contribute to the formation and maintenance of the Yellow
Sea Bottom Cold Water Mass (YBCWM), a distinct water mass in this marginal sea.

Over the past few decades, rising ocean temperatures have led to a significant increase in sea surface temperature (SST) in
the NWP and its marginal seas, exceeding the global average (Belkin, 2009). Additionally, the magnitude and frequency of
extreme climate events, such as marine heatwaves and cold surges, have increased markedly in the region (Horton et al.,
2015; Oliver et al., 2018; Tan and Cai, 2018; Yamaguchi et al., 2019; Yeo and Ha, 2019; Lee et al., 2020). For example, in
July 2021, the NWP experienced a record-breaking marine heatwave, with SST anomalies exceeding 3℃ in parts of the
East/Japan Sea and the Sea of Okhotsk compared to the 1982–2011 baseline period (Li et al., 2023). Furthermore, the
Kuroshio Current has intensified amid long-term oceanic warming trends (Chen et al., 2019), and its eastward inertial
extension, the Kuroshio Extension, has shifted northward, leading to substantial SST increases in surrounding oceans
(Kawakami et al., 2023).

To enable rapid responses to extreme marine events and accidents, as well as to address changes in oceanic conditions
affecting physical properties and marine ecosystems, the Korea Institute of Ocean Science and Technology (KIOST)
developed the high-resolution regional ocean model for the NWP, the Korea Operational Oceanographic System–Ocean
Predictability Experiment for the Marine Environment (KOOS-OPEM). The initial version of KOOS-OPEM (Kim et al.,
2009; Park et al., 2010) was a regional ocean circulation model for the East/Japan Sea, based on Modular Ocean Model



version 3 (MOM3; Pacanowski and Griffies, 1999), with a horizontal resolution of 1/10°. To improve the scientific understanding of the NWP and its marginal seas, KOOS-OPEM underwent multiple enhancements, including an increase in horizontal resolution to 1/24° to resolve the first baroclinic Rossby radius in most regions (Hallberg, 2013), expansion of the model domain, an upgrade from MOM3 to MOM5 (Griffies, 2012), and the incorporation of a data assimilation system based on Ensemble Optimal Interpolation (Kim et al., 2015).


**Figure 1. Bottom topography and ocean currents in the Northwest Pacific. (a) Full-region view and (b) zoomed-in view of the marginal seas, including the Yellow Sea, the East China Sea, and the East/Japan Sea, based on data from Park et al. (2013). Red arrows indicate warm currents, while blue arrows represent cold currents. (c) Five distinct regions used for regional temperature and salinity analysis: the open ocean of the Northwest Pacific (orange), the Kuroshio and its Extension (red), and three major**




**marginal seas—Yellow Sea (green), East/Japan Sea (blue), and the Sea of Okhotsk (purple). The Kuroshio and its exntension encompasses areas influenced by the Kuroshio Current and its extension, where the climatological surface current speed in GLORYS12 exceeds 0.3 m/s.**


Following these improvements, numerous studies have utilized KOOS-OPEM. Kim et al. (2021) applied an early version of the model to investigate the formation, variability, and pathways of intermediate waters in the East/Japan Sea. Yoon et al. (2022) explored the mechanisms driving summer phytoplankton blooms in Korean coastal waters using model outputs. Chang et al. (2023) assessed the contribution of satellite and in situ temperature observations to high-resolution regional

ocean modeling. Additionally, Chang et al. (2024) developed and evaluated a high-resolution regional ocean reanalysis for the NWP, while Jin et al. (2024) examined a 10-day ocean prediction system using KOOS-OPEM, which is operated weekly by KIOST, comparing it with other analysis and forecast fields.

In this study, we updated the base model of KOOS-OPEM to MOM6 (Adcroft et al., 2019), the latest version of MOM, to enhance its capabilities. MOM6 introduces a significantly different algorithm compared to previous versions (up to MOM5)

and offers substantial improvements in computational efficiency and stability. A key advancement is its use of vertical Lagrangian remapping (Griffies et al., 2020), a variant of the Arbitrary Lagrangian–Eulerian (ALE) algorithm, which allows for the implementation of various vertical coordinate systems, including geopotential (z or z*), isopycnal, terrain-following, or hybrid/user-defined coordinates. Additionally, newly developed open boundary conditions and improved regional modeling capabilities in MOM6 facilitate its effective use in regional ocean models.

Several recent studies have successfully implemented MOM6 in regional ocean modeling applications. Ross et al. (2023) conducted a hindcast simulation using MOM6 with the Sea Ice Simulator version 2 (SIS2) and the Carbon, Ocean Biogeochemistry, and Lower Trophics (COBALT; Stock et al., 2020) biogeochemistry model for the Northwest Atlantic from 1993 to 2019. Their simulation demonstrated excellent performance in reproducing physical properties, such as SST and Gulf Stream dynamics, while also exhibiting notable skill in modeling tidal behaviors and complex biogeochemical

processes. Seijo-Ellis et al. (2024) applied MOM6 in a high-resolution (1/12°) regional ocean modeling study of the Caribbean (CARIB12), effectively capturing the region's mesoscale variability, including eddy activity and the dynamics of major currents such as the Caribbean Current and the Loop Current. Furthermore, Drenkard et al. (submitted) implemented MOM6-COBALT for the Northeastern Pacific (NEP10k), covering the region from the Chukchi Sea to the Baja California Peninsula at a 10 km horizontal resolution. Their simulations successfully replicated key ecosystem-relevant properties,

including temperature, salinity, nutrient distributions, and chlorophyll concentrations, highlighting the model's capability to provide regionally tailored projections and support marine resource management. Additionally, Liao et al. (submitted) introduced MOM6-COBALT-IND12, a coupled physical-biogeochemical model for the northern Indian Ocean, which successfully represented monsoon-driven variability, coastal upwelling, and key ecosystem dynamics.

Spurious mixing in ocean models is a major concern, as it introduces an unphysical process that unintentionally increases

total mixing beyond the prescribed and parameterized levels (Griffies et al., 2000; Gibson et al., 2017). Consequently,



minimizing spurious mixing is a key focus in model development and configuration, with the choice of the vertical coordinate system playing a crucial role in determining its magnitude. The z* coordinate system (Adcroft and Campin, 2004, hereafter referred to as ZSTAR), as used in MOM5 and Ross et al. (2023), closely resembles the geopotential coordinate by scaling the vertical coordinate proportionally to sea surface height (SSH), allowing the upper ocean layers to remain thin.

This characteristic makes it particularly effective for capturing detailed processes in the ocean's mixed layer. Hybrid coordinates combine the advantages of different vertical coordinate systems to optimize model performance. A hybrid z*-isopycnal coordinate system (hereafter referred to as HYBRID), motivated by Bleck (2002), employs isopycnal coordinates in the ocean interior, where stratification is prominent, and ZSTAR coordinates in the unstratified mixed-layer regions. The HYBRID approach leverages the benefits of z-level coordinates for high resolution in the upper ocean while using isopycnal

coordinates in the deep ocean to minimize diapycnal mixing.

In this study, we conducted sensitivity experiments to compare the performance of the HYBRID and ZSTAR coordinate systems in a regional ocean model using the next-generation KOOS-OPEM (OPEM-MOM6). Although MOM6 can employ terrain-following coordinate systems, which are commonly used in regional ocean models, we excluded this option due to its poor performance associated with pressure gradient errors in regions with steep topography, such as the East/Japan Sea

(Haney, 1991; Beckmann and Haidvogel, 1993; Mellor et al., 1994; Chu and Fan, 1997; Mellor et al., 1998; Ezer et al., 2002). Therefore, our focus was on assessing the HYBRID and ZSTAR coordinate systems. Our primary objective was to evaluate how these systems influence the model's ability to capture key oceanographic features, processes, and dynamics through a quantitative analysis of their effects.

Section 2 describes the model configuration, including the implementation of different vertical coordinate systems, along

with the observational and reanalysis datasets used for evaluation. Section 3 presents the results of the sensitivity experiments, comparing the performance of the HYBRID and ZSTAR configurations in reproducing key oceanographic features. Finally, Section 4 provides a summary of the findings and discusses potential improvements for optimizing vertical coordinate representation in regional ocean modeling.

## 2 Methods

### 2.1 Model configuration

OPEM-MOM6 incorporates coupled model components using MOM6 for ocean physics and SIS2 for sea ice dynamics. The Arakawa-C grid system (Arakawa and Lamb, 1977) with $1704 \times 1392$ tracer points was used to solve the primitive equations under the Boussinesq and hydrostatic approximations. The model domain encompassed the NWP region, spanning from 99°E to 170°E and 5°N to 63°N. The model had a horizontal resolution of 1/24° in both longitude and latitude, equivalent to

approximately 4 km. The bathymetry data were constructed by integrating the General Bathymetric Chart of the Oceans (GEBCO) 2024 and the Korbathy dataset, a regional bathymetry dataset for the Korean Peninsula (Seo, 2008). The minimum





bathymetric depth was set to 10 m to account for tidal variations, as wetting and drying were not employed, while the maximum depth was limited to 5,000 m to enhance computational efficiency.

To integrate the ocean model forward in time, a split explicit method (Hallberg, 1997; Hallberg and Adcroft, 2009) was employed, efficiently separating the handling of fast and slow processes. The baroclinic time step was set to 300 seconds, while the barotropic time step varied and was determined as the largest integer fraction of the baroclinic time step required for stability. A longer time step of 900 seconds was applied for thermodynamic calculations.

In this study, two vertical coordinate systems, HYBRID and ZSTAR, were configured with 75 layers. Both configurations featured the finest vertical resolution near the surface, with a layer thickness of 2 meters extending to a depth of 14 meters in the ZSTAR space. In ZSTAR, the layer thickness gradually increased with depth, reaching a maximum of 349.43 meters just above the deepest model depth of 5,000 meters. In the HYBRID configuration, ZSTAR was used to effectively resolve the mixed layer in unstratified regions, providing high resolution where vertical mixing and surface interactions were most significant. Below the mixed layer, isopycnal coordinates were employed to minimize spurious diapycnal mixing and accurately represent the stratified conditions found in deeper waters. The transition depth between the isopycnal and ZSTAR coordinates deepened toward higher latitudes (Adcroft et al., 2019). In MOM6, HYBRID assigned a target density referenced to 2,000 dbar for each interface. The target density ranged between $\sigma_2$ 10.00 and 38.00 kg m$^{-3}$, as used by Adcroft et al. (2019). The choice of a 2,000 dbar reference pressure is widely accepted, as it balances monotonicity in near-surface waters with stability in the deep ocean (Megann, 2018) and maximizes the neutrality of isopycnal surfaces (McDougall and Jackett, 2005). However, variations in the vertical density distribution across global and regional scales led to inefficient resolution use, particularly in weakly stratified regions. To address this, 1% of the compressibility was artificially retained but centered at 2,000 dbar when generating the vertical grid at each time step (Adcroft et al., 2019).

The physical subgrid-scale parameterization settings followed those of Adcroft et al. (2019) and Ross et al. (2023). The energetic planetary boundary layer scheme developed by Reichl and Hallberg (2018), with updates accounting for Langmuir turbulence (Reichl and Li, 2019), was employed to parameterize the planetary boundary layer. To parameterize mixed-layer restratification by sub-mesoscale eddies, the scheme proposed by Fox-Kemper et al. (2011) was used, with a frontal-length scale of 1,500 meters applied for the upscaling of buoyancy gradients. A biharmonic form of horizontal viscosity was used in these simulations. The viscosity was calculated as the maximum value between a biharmonic Smagorinsky viscosity (Griffies and Hallberg, 2000) and a predefined fixed viscosity expressed as $u_4 \Delta x^3$, where $u_4$ represents a velocity scale and $\Delta x$ denotes the local grid spacing. The velocity scale was set to 0.01 m/s, with a Smagorinsky coefficient of 0.015. Shear-driven turbulence mixing was parameterized by Jackson et al. (2008). Table 1 provides a summary of the model configurations and parameters.




**Table 1. Summary of model configuration and parameters for each experiment**

| Parameter | HYBRID | ZSTAR | Reference |
|---|---|---|---|
| **Vertical coordinate** | 75-hybrid (z*-isopycnal coordinate) | 75-Z* coordinate | |
| **Horizontal resolution** | 1/24° | | |
| **Domain** | 99°E ~ 170°E / 5°N ~ 63°N (1704 x 1392 tracer points) | | |
| **Time stepping** | | | |
| Baroclinic | 300 s | | |
| Thermodynamics | 900 s | | |
| **Tides** | | | |
| | 10 Tidal constituents (M2,S2,N2,K2,K1,O1,P1,Q1,MM,MF) | | |
| Tidal Potential | | | Egbert and Erofeva (2002) |
|   Explicit from TPXO | | | |
| **Open boundary condition** | | | |
|   Barotropic | Flather | | Flather (1976) |
|   Baroclinic | Orlanski (nudging timescale: 3d for inflow, 360d for outflow) | | Orlanski (1976) |
|   Tracer | Reservoirs with 9 km length scales | | |
| **Background kinematic viscosity** | $1.0 \times 10^{-6} m^2 s^{-1}$ | | |
| **Background diapycnal diffusivity** | $1.0 \times 10^{-6} m^2 s^{-1}$ | | |
| **Horizontal Viscosity** | Biharmonic (The maximum value between Smagorinsky and resolution-dependent viscosities ) | | |
|   Smagorinsky coefficient | 0.015 | | |
|   Resolution-dependent | $0.01 \, \Delta_x^3 \, m^4 s^{-1}$ | | |
| **Ocean boundary layer parameterization** | ePBL | | Reichl and Hallberg (2018) |
| **Mixed Layer re-stratification** | Front length scale = 1500 m | | Fox-Kemper et al. (2011) |



## 2.2 Model forcing and spin-up


Both configurations were forced using lateral open boundary conditions from the GLORYS12 reanalysis (Jean-Michel et al., 2021). The variables used for lateral boundary conditions included daily mean temperature, salinity, sea surface height (SSH), and ocean velocity. The model was forced by astronomical tidal potential forcing, with explicit tidal forcing from boundary conditions rather than parameterized tidal mixing. Tidal harmonics, including four semidiurnal (M2, S2, N2, and K2), four diurnal (K1, O1, P1, and Q1), and two long-period (Mm and Mf) constituents from the TPXO9 v1 dataset (Egbert and Erofeeva, 2002), were applied to impose tidal variations in sea level and velocity on the subtidal boundary data. These 10 constituents were also applied as body forces in the momentum equations to simulate astronomical tidal forcing across the domain. Combined tidal and subtidal sea levels, along with barotropic velocities, were prescribed using the radiation boundary conditions described by Flather (1976). For baroclinic flow, a radiation scheme based on Orlanski (1976) was applied, incorporating nudging toward external data following the approach outlined by Marchesiello et al. (2001). Inflow boundary velocities were strongly constrained with a 3-day nudging timescale, while outflow velocities were weakly adjusted using a 360-day timescale. Temperature and salinity at the boundaries were managed using a reservoir scheme, which adapted boundary conditions based on the internal model state for outflow and external forcing for inflow. The reservoir length scale was set to 9 km.



Surface fluxes between the ocean and atmosphere were derived from hourly ERA5 reanalysis data (Hersbach et al., 2020), including variables such as 2-m air temperature, specific humidity, surface net solar and thermal radiation, mean sea level pressure, total cloud cover, 10-m wind velocity, and precipitation, using the bulk formula from Large and Yeager (2004). River discharge was prescribed using the GloFAS reanalysis version 3.1 (Alfieri et al., 2020). Following the approach of Ross et al. (2023), river discharge was mapped onto the MOM6 grid by identifying coastal outlet points and assigning streamflow to the nearest ocean grid cell using a local drain direction map. A comparison with observations from Datong station revealed that the GloFAS dataset overestimated the Yangtze River discharge. To correct this bias, GloFAS discharge data were adjusted using a bias correction based on the monthly climatological runoff ratio. Freshwater, with zero salinity and a temperature equal to the surface temperature of the discharge grid cell, was added at the surface. Additionally, turbulent kinetic energy was introduced to mix the water column up to a depth of 5 m at discharge points.



Both configurations were initialized using temperature and salinity fields from GLORYS12, which were interpolated to the model grid from January 1, 1993. The spin-up simulation was run for 10 years (1993–2002) using time-varying open boundary and surface atmospheric forcing data. Following the spin-up period, hindcast simulations were performed for 2003–2012 to capture and analyze oceanographic conditions and dynamics. This approach ensured that the models were sufficiently spun up and provided a reliable representation of ocean conditions during the specified period.




## 2.3 Evaluation

The performance of each configuration was evaluated using observational data and physical reanalysis datasets that assimilated observations. For statistical evaluation, Iris v3.1.0 (Hattersley et al., 2023), a Python package for analyzing and visualizing multi-dimensional meteorological and oceanographic datasets, was used to compare both configurations with the reference dataset. For visualization, Cartopy (Met Office, 2022) was employed to represent geographic features, while Iris was utilized to display the computed variable distributions.

Since reference datasets generally had a lower spatial resolution than the model outputs, model outputs from both configurations were conservatively re-gridded onto the coarser-resolution reference dataset grid using Iris before conducting statistical analysis. The statistical evaluation included spatial mean bias (Bias), root mean squared error (RMSE), median absolute error (MedAE), and the Pearson correlation coefficient (Corr). Bias indicated whether the model systematically overestimated or underestimated values. RMSE quantified the overall discrepancy between the model and reference datasets by measuring squared differences, making it sensitive to large errors. In contrast, MedAE provided a robust measure of error by calculating the median of absolute differences, reducing the influence of outliers compared to RMSE. Corr measured similarity in spatial or temporal patterns, ranging from -1 (inverse correlation) to 1 (perfect correlation), independent of magnitude differences.

To compare SST from both configurations, the NOAA 1/4° daily Optimum Interpolation Sea Surface Temperature (OISST) dataset, which interpolates and extrapolates observations from satellites, Argo floats, ships, and buoys, was used to evaluate SST performance in the experiments. Sea surface salinity (SSS) validation was conducted using the GLORYS12 reanalysis dataset. Although observational data for SSS are limited, alternative datasets, such as the CMEMS Multi-Observation Global Ocean Sea Surface Salinity product, have been used in previous studies (e.g., Seijo-Ellis et al., 2024). However, the CMEMS product relies on climatological data for coastal areas, where observational coverage is sparse and uncertainty is high. Specifically, the coastal dataset incorporates pseudo-observations derived from climatological backgrounds within 200 km of the coast, as described in CMEMS documentation. In contrast, GLORYS12 has been shown to exhibit relatively low bias around the Korean Peninsula when compared with in-situ observations (Chang et al., 2023), and also demonstrated reasonable temporal variability when compared with SSS time series from the IEODO Ocean Research Station, located in East China Sea (not shown here). Based on these findings, GLORYS12 was deemed a suitable reference for SSS validation in this study.

The marginal seas of the NWP are heavily influenced by significant freshwater discharge from the Yangtze and Yellow Rivers, which creates extensive low-salinity areas that play a critical role in regional salinity distribution and stratification. Therefore, validating SSS in coastal regions is essential for accurately assessing model performance. Given its consistency in representing salinity distributions across both coastal and open-ocean areas, the GLORYS12 reanalysis dataset was selected as the reference.





The boreal winter and summer mixed layer depth (MLD) was compared to the long-term MLD climatology constructed from World Ocean Database and Argo profiles (de Boyer Montégut, 2023). In this dataset, MLD is calculated using the $\Delta 0.03$ kg m$^{-3}$ density criterion relative to surface density. The MOM6 diagnostic MLD_003 was used for validation, defining MLD as the depth where potential density exceeds the density at 10 m by 0.03 kg m$^{-3}$.

To evaluate sea surface height (SSH) variability, monthly absolute dynamic topography (ADT) data from CMEMS were used. These data, produced by merging SSH observations from various satellite altimetry sources, have a horizontal resolution of 0.25°. Additionally, GLORYS12 was used to validate surface current speed and eddy kinetic energy (EKE) in the model simulations. The EKE for each experiment was calculated by interpolating velocity data onto the AVISO geostrophic current data grid and applying the following equation:

$$EKE = \frac{1}{2}(u'^2 + v'^2)$$

where $u'$ and $v'$ represent deviations of the zonal and meridional velocity components from their respective means.

To assess the model's ability to reproduce tides, the tidal amplitudes and phases of the M2 (semidiurnal) and K1 (diurnal) constituents were calculated using hourly SSH output with the UTide Python package (Codiga et al., 2011). The results were then compared with TPXO9, which served as the tidal boundary condition.

Additionally, two reanalysis datasets, GLORYS12 and KOOS-OPEM ReAnalysis 2022 (K-ORA22; Chang et al., 2024), were used to evaluate the performance of each experiment in reproducing vertical temperature and salinity structures. A comparative evaluation by Chang et al. (2023) assessed multiple NWP reanalysis datasets, finding that GLORYS12 exhibited the best performance in reproducing temperature, salinity, and large-scale/mesoscale variability. Meanwhile, K-ORA22 demonstrated superior performance in representing marginal seas, particularly excelling in the Yellow Sea. Given these complementary strengths, GLORYS12 and K-ORA22 were selected as reference datasets for evaluating vertical temperature and salinity structures.

To compare regional variations in temperature and salinity between the two configurations, the analysis was divided into five distinct regions, as shown in Fig. 1c:

1. Open ocean area of the NWP,
2. Kuroshio and its Extension,
3. Sea of Okhotsk,
4. East/Japan Sea, and
5. Yellow Sea

The K-KE regions were defined not only based on areas where the climatological surface current speed in GLORYS12 exceeds 0.3 m/s but also by including adjacent regions dynamically affected by the K-KE. A summary of the parameters and datasets used in the evaluation is provided in Table 2.

**Table 2. Summary of parameters and dataset used in the evaluation**





| Parameter | Time sampling | Horizontal resolution | Dataset (Reference) |
|---|---|---|---|
| Sea Surface Temperature | Seasonal mean Climatology | 1/4° | OISST v2 (Huang et al., 2021) |
| Sea Surface Salinity | Seasonal mean Climatology | 1/12° | GLORYS12 (Jean-Michel et al., 2021) |
| Mixed Layer Depth | Seasonal mean Climatology | 1° | de Boyer Montégut (2024) |
| Sea Surface Height | Annual mean climatology | 1/4° | Gridded Sea surface height (CMEMS) |
| Large scale and mesoscale variability | Monthly | 1/4° | Gridded Sea surface height (CMEMS) |
| Current speed | Annual mean climatology | 1/12° | GLORYS12 (Jean-Michel et al., 2021) |
| Vertical temperature/salinity | Annual mean climatology | 1/12° | GLORYS12 (Jean-Michel et al., 2021) |
| | | 1/24° | K-ORA22 (Chang et al., 2024) |
| Volume transport (Korea/Tsushima strait) | Monthly mean climatology | - | Shin et al. (2022) |
| Volume transport (Tokara, Tsugaru, and Soya) | Annual mean climatology | - | Wei et al. (2013) Han et al. (2016) Ohshima and Kuga (2023) |
| Tidal amplitude and phase (M2 and K1) | Hourly mean | 1/6° | TPXO9 (Egbert and Erofeeva, 2002) |

## 3 Results

### 3.1 Near-surface physical ocean properties

Figure 2 compares the SST distributions for boreal winter (DJF) between the OISST observational dataset and simulations from the HYBRID and ZSTAR configurations. The winter mean SST distributions from OISST and both model configurations exhibited strong agreement in both spatial structure and magnitude across the NWP. Biases relative to OISST were low, with HYBRID showing a bias of -0.02℃ and ZSTAR showing a bias of 0.15℃, indicating overall consistency with observations. Despite this broad agreement, regional biases were evident. In higher latitudes north of 45°N, both configurations exhibited a moderate cold bias of approximately -0.8℃, with ZSTAR displaying a more pronounced cold bias compared to HYBRID. Warm biases were observed in the Kuroshio and the Kuroshio-Oyashio transition zone, reaching up to 2.0℃ in both configurations. However, this warm bias was more prominent in the HYBRID configuration, particularly in the Kuroshio-Oyashio transition zone. Additionally, the HYBRID configuration exhibited a warm bias in the South China



Sea, whereas the ZSTAR configuration demonstrated a more substantial warm bias exceeding 3.0℃ in the East/Japan Sea, which was notably larger than in HYBRID.

Statistical metrics further supported the agreement between the models and OISST. Both configurations achieved high spatial correlations of 0.98, with RMSE values of 0.71℃ for HYBRID and 0.74℃ for ZSTAR. Median absolute errors

(MedAE) were 0.33℃ for HYBRID and 0.31℃ for ZSTAR, reflecting similar levels of accuracy in representing SST variability across the region.

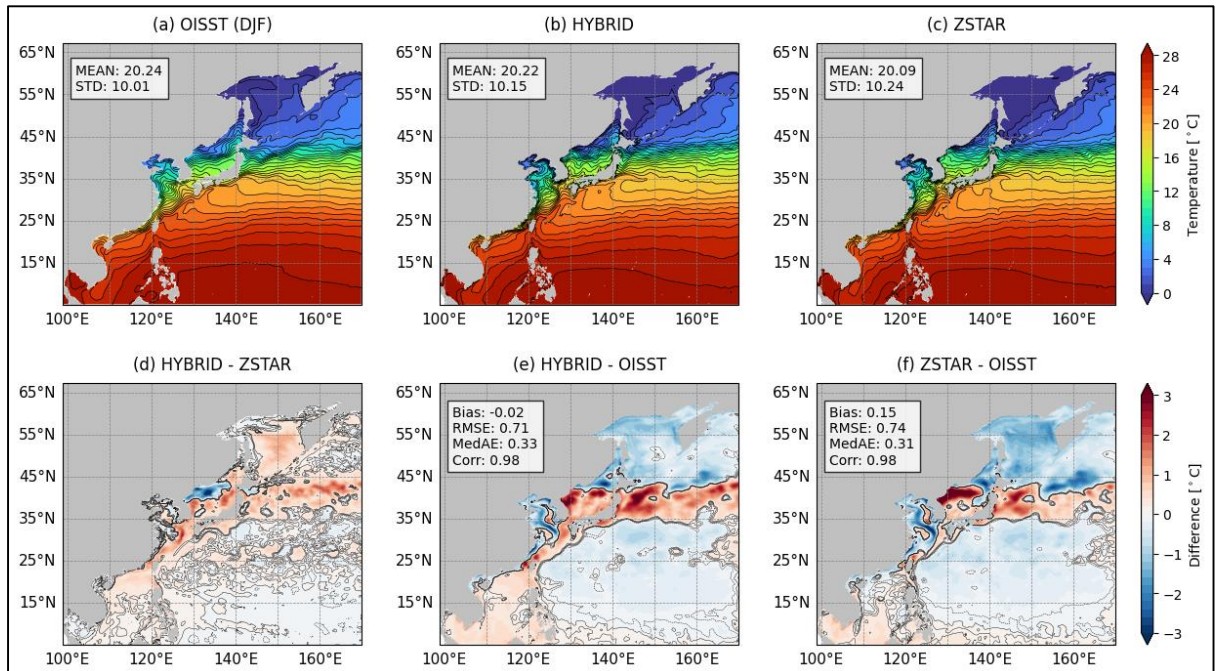

**Figure 2. Boreal winter (DJF) mean sea surface temperature (SST) distributions from OISST observations and HYBRID and**
295 **ZSTAR simulations. (a–c) Spatial SST distributions with corresponding means and standard deviations (STD). (d) Differences**
**between HYBRID and ZSTAR. (e, f) Biases relative to OISST, including mean bias, root mean squared error (RMSE), median**
**absolute error (MedAE), and correlation (Corr). Contour lines in (d–f) indicate SST biases ranging from -0.1 to 0.1 ℃ at 0.1 ℃**
**intervals.**

Figure 3 presents a comparison of SST distributions for boreal summer (JJA) between OISST observations and simulations from the HYBRID and ZSTAR configurations. Both configurations exhibited similar performance, with RMSE values of 0.70℃ for HYBRID and 0.66℃ for ZSTAR, and both maintained high correlations with OISST (Corr = 0.98), consistent with their performance in winter. However, unlike in winter, both models exhibited warm biases, particularly in the Yellow Sea and high-latitude regions, where biases of approximately 1℃ were observed. In contrast, biases in the open ocean of the
NWP remained relatively low, typically below -0.3℃. ZSTAR exhibited larger temperature biases than HYBRID in the



Yellow Sea and East/Japan Sea, while HYBRID exhibited stronger warm biases in the South China Sea and Kuroshio-Oyashio transition zone compared to ZSTAR.

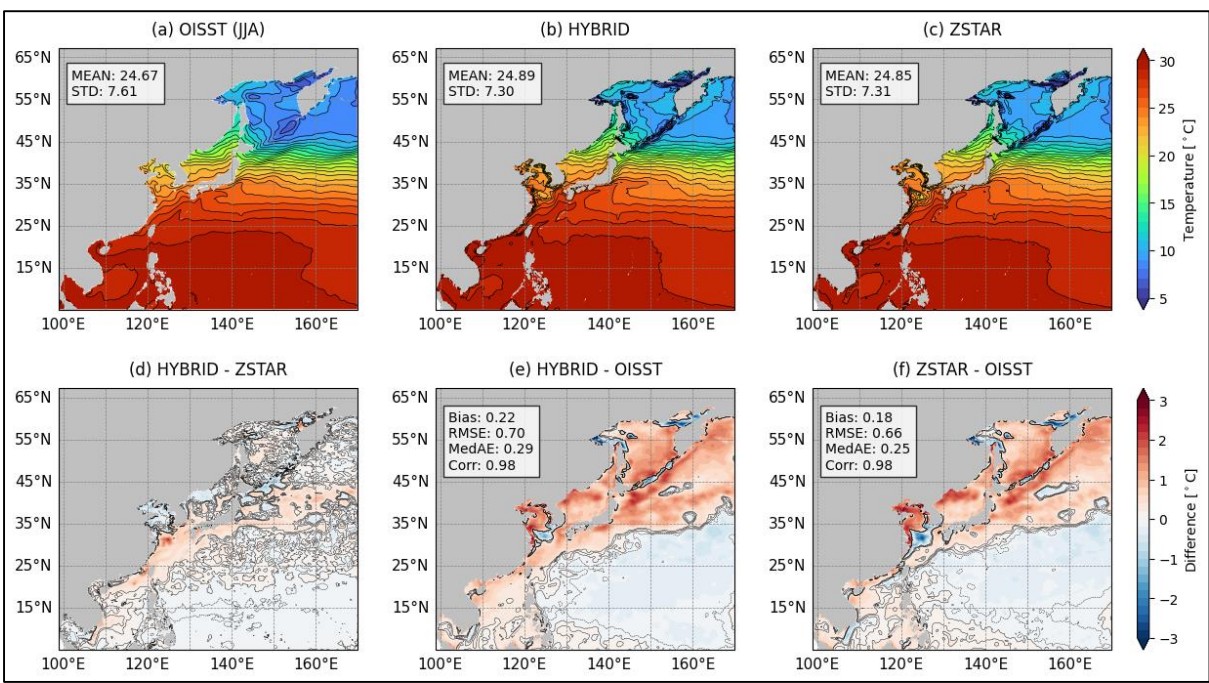

**Figure 3. Boreal summer (JJA) mean sea surface temperature (SST) distributions from OISST observations and HYBRID and**
**ZSTAR simulations. (a–c) Spatial SST distributions with corresponding means and standard deviations (STD). (d) Differences**
**between HYBRID and ZSTAR. (e, f) Biases relative to OISST, including mean bias, RMSE, MedAE, and Corr. Contour lines in**
**(d–f) indicate SST biases ranging from -0.1 to 0.1 °C at 0.1 °C intervals.**

The simulated mean SSS values for both HYBRID and ZSTAR configurations (33.99 psu) aligned closely with the GLORYS12 reference mean (33.92 psu), demonstrating good agreement in the large-scale salinity distribution across the NWP (Fig. 4). However, the standard deviations revealed slightly higher variability in the models, with values of 1.53 psu for HYBRID and 1.61 psu for ZSTAR, compared to 1.32 psu in GLORYS12. This indicated that while both models captured regional salinity gradients reasonably well, they tended to overestimate variability.

Regional biases were evident in specific areas. In the Yellow Sea, both configurations exhibited a pronounced negative bias exceeding -1.0 psu, suggesting that excessive freshwater discharge from nearby rivers led to an overestimation of low-salinity water in this region. Conversely, positive biases dominated in the South China Sea, the southeastern Chinese coast, the northern East/Japan Sea, and Sea of Okhotsk, with values exceeding 1.0 psu in some locations. Across the open ocean, as well as in the Yellow Sea and South China Sea, ZSTAR showed slightly larger positive biases than HYBRID, while HYBRID exhibited larger biases than ZSTAR in the Sea of Okhotsk and the Kuroshio-Oyashio transition zone. A quantitative comparison with GLORYS12 highlighted differences in model performance. HYBRID achieved a lower RMSE



of 0.72 psu compared to 0.80 psu for ZSTAR, suggesting better overall agreement with the reference dataset. Additionally, HYBRID showed a marginally higher spatial correlation with GLORYS12 (0.82) compared to ZSTAR (0.81).

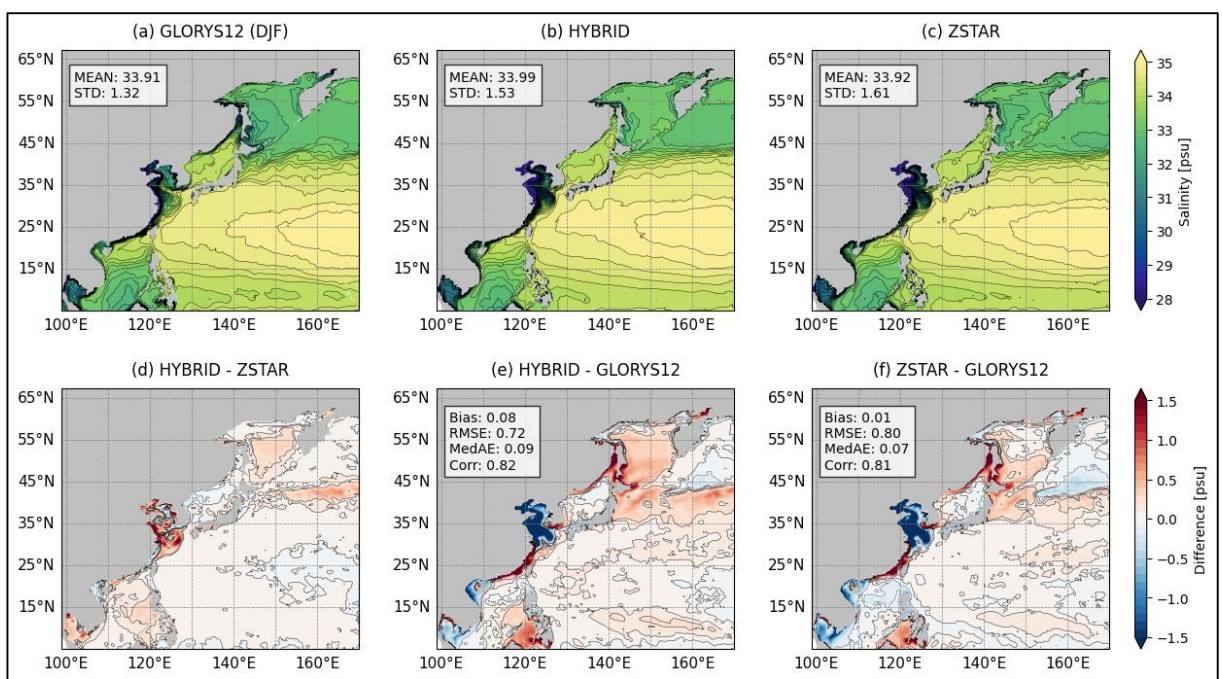

**Figure 4. Boreal winter (DJF) mean sea surface salinity (SSS) distributions from the GLORYS12 reanalysis and HYBRID and ZSTAR simulations. (a–c) Spatial SSS distributions with corresponding means and standard deviations (STD). (d) Differences between HYBRID and ZSTAR. (e, f) Biases relative to GLORYS12, including mean bias, RMSE, MedAE, and Corr. Contour lines in (d–f) indicate SSS biases ranging from -0.1 to 0.1 psu at 0.1 psu intervals.**

During summer (JJA), the SSS distribution remained similar to winter, with both configurations aligning well with GLORYS12 but showing slightly higher variability (Fig. S1). The negative bias in the Yellow Sea intensified, exceeding -1.0 psu, while regional bias patterns persisted, with ZSTAR showing stronger positive biases in the open ocean and HYBRID exhibiting more negative biases in the Kuroshio-Oyashio transition zone and Sea of Okhotsk. This suggested that, despite applying bias corrections for Yangtze River discharge, river discharge forcing in both configurations may have still been overestimated for other rivers, such as the Yellow River. Therefore, further investigation into discharge from other major rivers is necessary, comparing them with observational datasets for potential corrections. Additionally, shifting the river mouths slightly inland could help reduce the excessive influence of river discharge on coastal salinity distributions. In other regions, ZSTAR showed stronger positive biases in the open ocean of the NWP compared to HYBRID, while HYBRID exhibited more pronounced negative biases than ZSTAR in the Kuroshio-Oyashio transition zone and Sea of Okhotsk, with some values exceeding 1.0 psu.



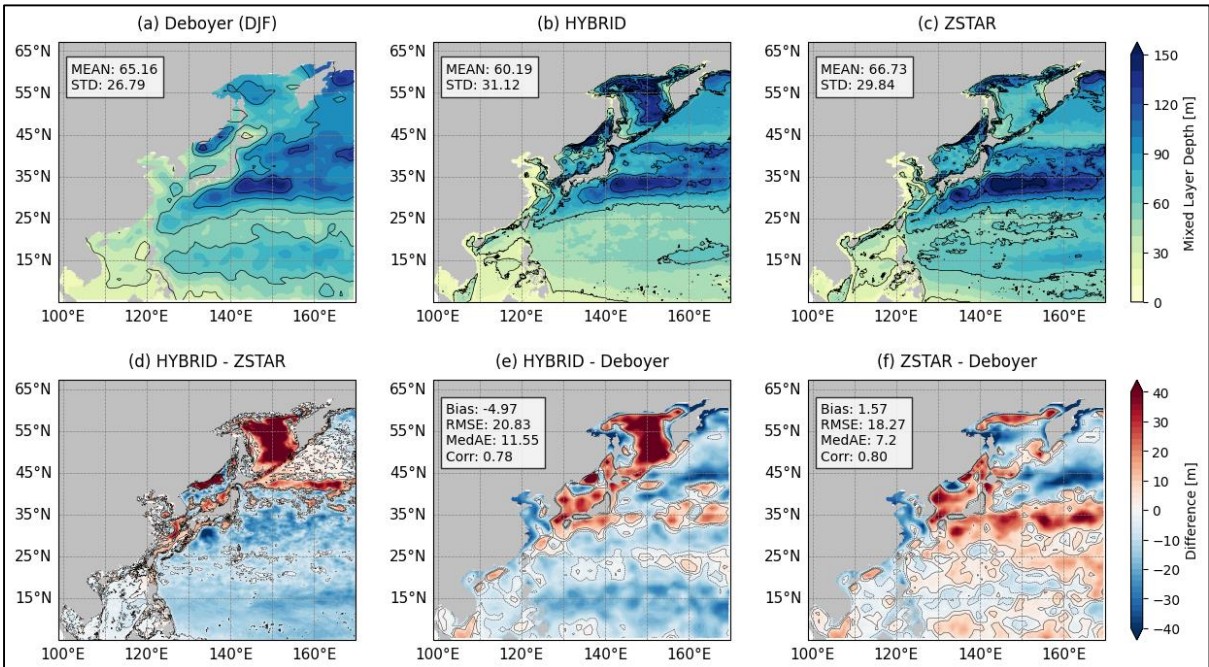

**Figure 5. Boreal winter (DJF) mean mixed layer depth (MLD) distributions from de Boyer Montégut and HYBRID and ZSTAR simulations. (a–c) Spatial MLD distributions with corresponding means and standard deviations (STD). (d) Differences between HYBRID and ZSTAR. (e, f) Biases relative to de Boyer Montégut, including mean bias, RMSE, MedAE, and Corr. Contour lines**
**in (d–f) indicate MLD biases ranging from -0.1 to 0.1 m at 0.1 m intervals.**

Figure 5 compares the MLD distributions for boreal winter (DJF) between the estimates of de Boyer Montégut et al. (2024) and simulations from the HYBRID and ZSTAR configurations. ZSTAR generally simulated a deeper MLD, particularly in regions influenced by western boundary currents, such as the Kuroshio and EKWC, where it overestimated MLD by

355 approximately 20 m relative to the reference, with a mean bias of 1.57 m and an RMSE of 18.27 m. HYBRID exhibited a larger negative bias of about 15 m compared to ZSTAR in the open ocean of the NWP. In addition, HYBRID showed a significant negative bias exceeding 30 m in Sea of Okhotsk. The RMSE for HYBRID was 20.83 m, higher than that for ZSTAR (18.27 m), primarily due to the substantial bias in the sea of Okhotsk. This indicated that while both configurations showed high spatial correlations with the reference data, they exhibited tendencies to either overestimate or underestimate

MLD depths depending on the region, with ZSTAR particularly overestimating MLD in dynamic boundary current areas and HYBRID showing larger biases in high-latitude regions.

Figure S3 also presents the MLD distribution for boreal summer (JJA), when the mixed layer is generally shallower due to enhanced stratification. The estimated summer MLD was shallow across most of the domain (mean: 19.15 m), deepening in western boundary current regions. HYBRID underestimated the MLD (mean: 16.82 m), while ZSTAR slightly



overestimated it (mean: 19.57 m). ZSTAR showed a higher spatial correlation with the observation (0.91 vs. 0.83) and along
with lower RMSE (4.35 m vs. 5.08 m).

## 3.2 Upper-ocean circulation and variability

To ensure a consistent reference level when comparing SSH between the models and the Altimetry dataset, the mean
absolute difference between each model and Altimetry SSH was subtracted from the respective model outputs. Overall, both
the HYBRID and ZSTAR configurations exhibited SSH distributions that closely aligned with Altimetry, effectively
capturing the large-scale features of the region (Fig. 6). The standard deviation values of SSH variability, representing spatial
gradients, were 43.34 cm for HYBRID and 42.13 cm for ZSTAR, compared to 41.78 cm for Altimetry. These values
indicate that both configurations reproduced SSH gradients well, with HYBRID exhibiting slightly larger spatial variability
than ZSTAR. Both configurations showed similar SSH biases, with an overall underestimation south of Japan, where
Kuroshio recirculation occurs. However, HYBRID exhibited a higher SSH than ZSTAR in the Kuroshio and its Extension
region. The spatial correlation coefficients of 0.94 for both HYBRID and ZSTAR indicate strong agreement with the
reference dataset.

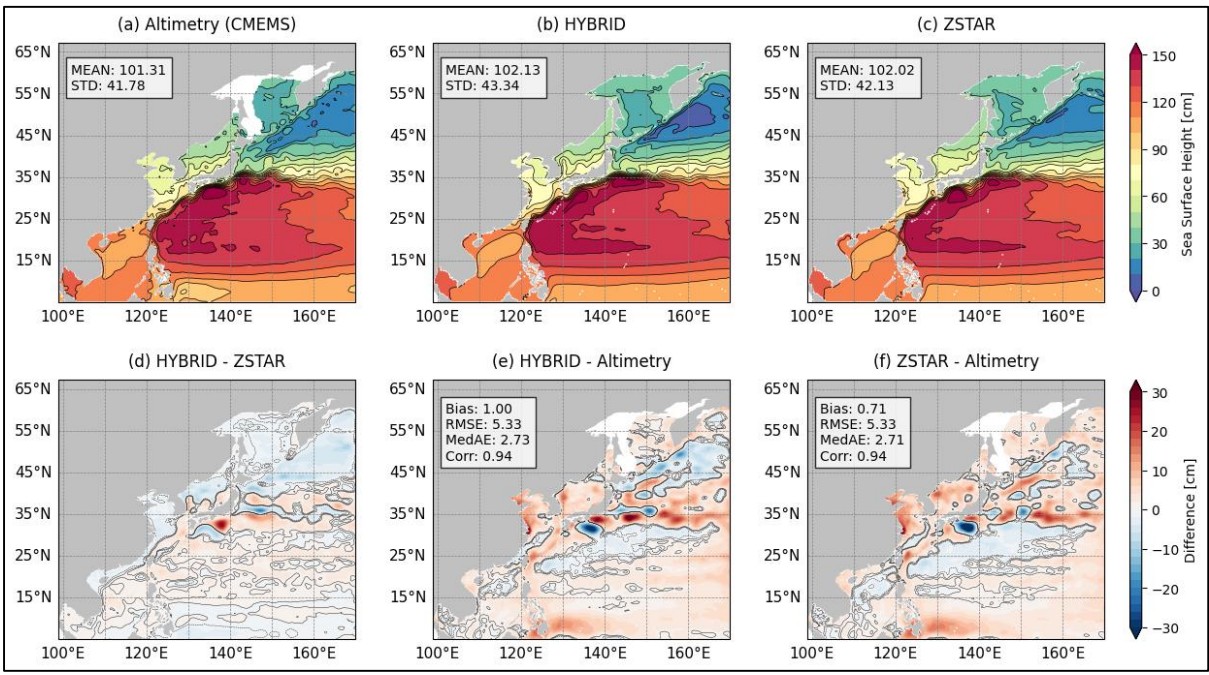

**Figure 6. Mean sea surface height (SSH) distributions from Altimetry data and HYBRID and ZSTAR simulations. (a–c) Spatial
SSH distributions with corresponding means and standard deviations (STD). (d) Differences between HYBRID and ZSTAR. (e, f)
Biases relative to Altimetry, including mean bias, RMSE, MedAE, and Corr. Contour lines in (d–f) indicate SSH biases ranging
from -1.0 to 1.0 cm at 1.0 cm intervals.**



As described by Qiu (2023) and Chang et al. (2024), SSH variability can be divided into large-scale and mesoscale components. This classification was based on a frequency spectrum analysis of Altimetry data, which revealed prominent peaks at both high and low frequencies, with a sharp decline to near zero at approximately two years. Accordingly, applying low-pass and high-pass filters allows for the separation of these components, effectively distinguishing large-scale ocean circulation, which evolves over interannual to decadal timescales, from high-frequency mesoscale eddy variability, which is characterized by shorter-lived fluctuations such as eddies and meanders.

Figure 7 compares the large-scale SSH variability of the HYBRID and ZSTAR configurations with Altimetry after applying a two-year low-pass filter. The root-mean-square (RMS) SSH variability was 2.81 cm for HYBRID and 2.69 cm for ZSTAR, both lower than the Altimetry reference value of 4.92 cm. The standard deviations for HYBRID and ZSTAR similarly indicated reduced variability compared to Altimetry, which had a standard deviation of 3.48 cm. Both models underestimated large-scale variability in key dynamic regions, including the North Equatorial Current, Kuroshio Current, and Kuroshio Extension. In particular, HYBRID exhibited a pronounced underestimation in the Kuroshio Extension region while overestimating variability in the Kuroshio Recirculation area and the East/Japan Sea. In terms of spatial correlation, ZSTAR achieved a higher correlation coefficient (0.66) with Altimetry than HYBRID (0.56), indicating better spatial agreement in large-scale variability patterns.

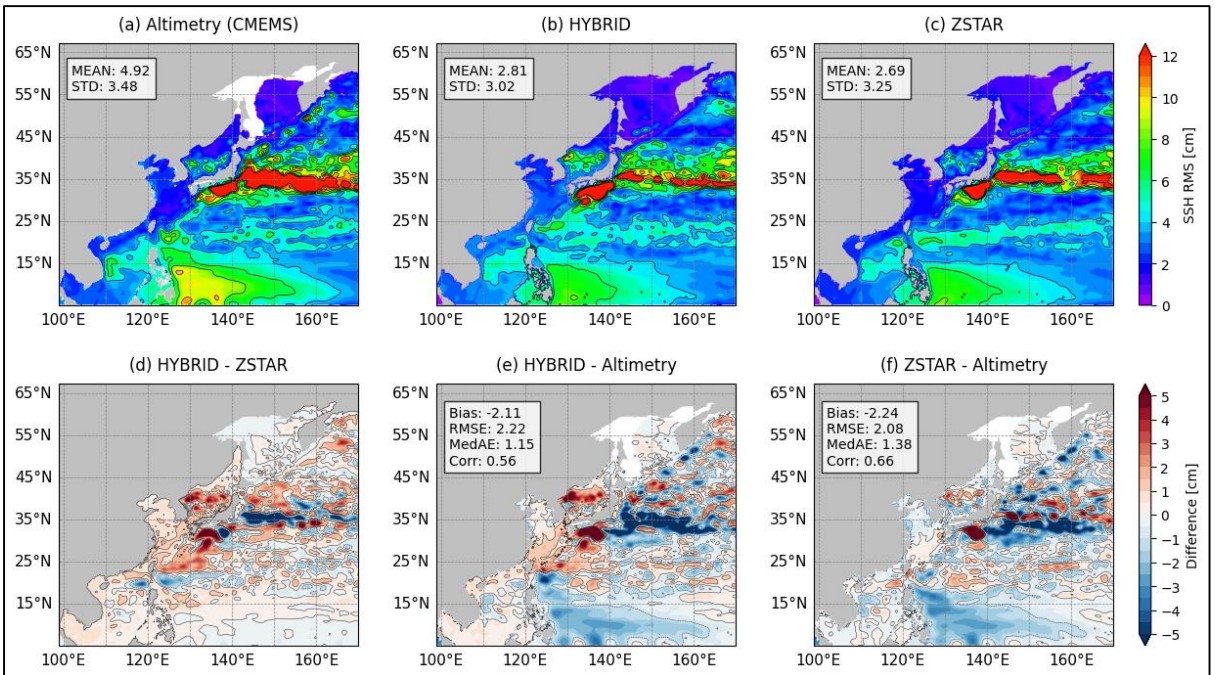

**Figure 7. Root-mean-square (RMS) sea surface height (SSH) variability from low-pass filtered Altimetry data and HYBRID and ZSTAR simulations. (a–c) Spatial RMS SSH distributions with corresponding means and standard deviations (STD). (d)**



**Differences between HYBRID and ZSTAR. (e, f) Biases relative to Altimetry, including mean bias, RMSE, MedAE, and Corr.**
**Contour lines in (d–f) indicate RMS SSH biases ranging from -1.0 to 1.0 cm at 1.0 cm intervals.**

Figure 8 illustrates the mesoscale variability of SSH after applying a high-pass filter, which extracts the high-frequency components associated with mesoscale eddies and smaller-scale oceanographic features. The mean RMS SSH variability was 4.64 cm for HYBRID and 4.41 cm for ZSTAR, both significantly lower than the Altimetry reference value of 8.54 cm.
Similarly, the standard deviations for HYBRID (4.56 cm) and ZSTAR (4.42 cm) were lower than those of Altimetry (4.76 cm). The discrepancies between both configurations and Altimetry were further reflected in the bias and RMSE values. HYBRID exhibited a bias of -3.90 cm and an RMSE of 2.55 cm, while ZSTAR had a bias of -4.05 cm and an RMSE of 2.64 cm. ZSTAR underestimated mesoscale variability in the open ocean of the NWP, whereas HYBRID showed a more pronounced underestimation in the Kuroshio and its Extension compared to ZSTAR. Additionally, both configurations had
low spatial correlations with Altimetry, with HYBRID achieving a correlation of 0.57 and ZSTAR a slightly higher correlation of 0.59. These results suggest that while both configurations captured some aspects of mesoscale variability, they tended to underestimate its intensity and struggled to fully resolve the finer-scale structures observed in the Altimetry data.

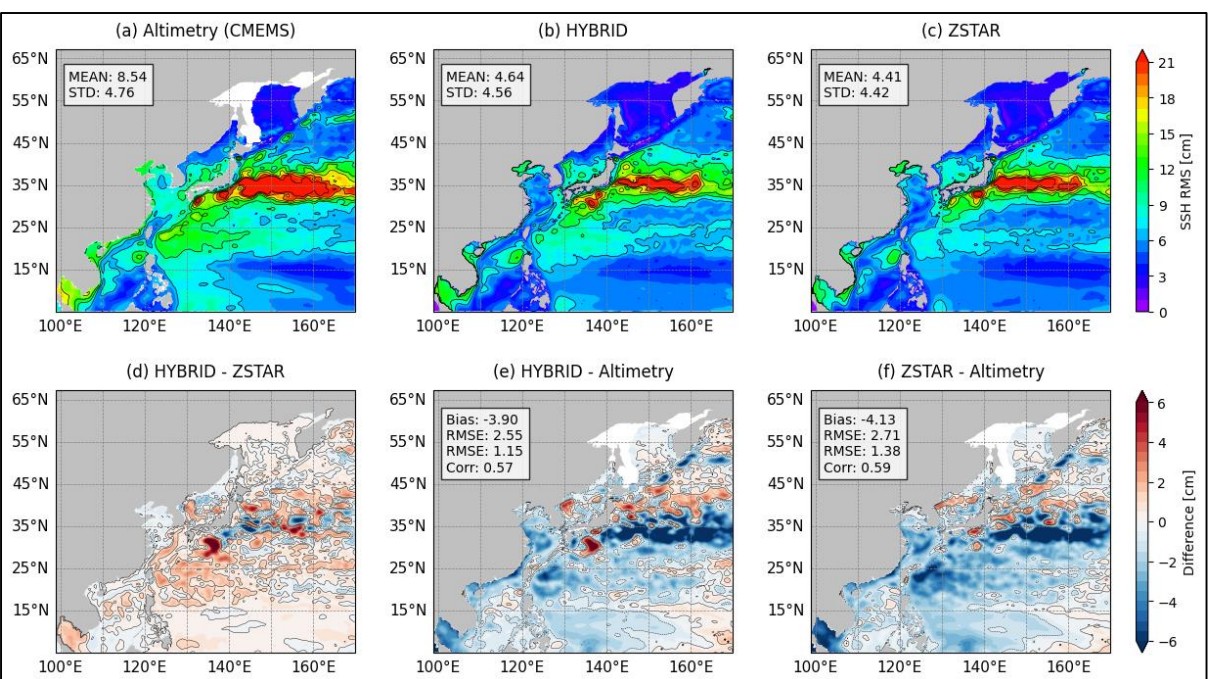

**Figure 8. Root-mean-square (RMS) sea surface height (SSH) variability from high-pass filtered Altimetry data and HYBRID and ZSTAR simulations. (a–c) Spatial RMS SSH distributions with corresponding means and standard deviations (STD). (d) Differences between HYBRID and ZSTAR. (e, f) Biases relative to Altimetry, including mean bias, RMSE, MedAE, and Corr. Contour lines in (d–f) indicate RMS SSH biases ranging from -1.0 to 1.0 cm at 1.0 cm intervals.**





Figure 9 compares the surface current speeds of the HYBRID and ZSTAR configurations with the GLORYS12 reanalysis. Both models effectively captured the complex ocean current systems in the NWP, including key currents such as the Kuroshio, Oyashio, and North Equatorial Current. However, both configurations tended to underestimate current speeds in regions influenced by these currents. Specifically, in areas such as the Kuroshio and its extension, the Subtropical Counter Current, the North Equatorial Current, and the South China Sea, both models underestimated current speeds compared to

GLORYS12. In contrast, both configurations overestimated current speeds in the East/Japan Sea. The East Korea Warm Current also exhibited an overshoot in both models, but differences existed between configurations. HYBRID simulated the East Korea Warm Current with an overshoot but captured the separation point more accurately than ZSTAR. In contrast, ZSTAR simulated the East Korea Warm Current separating farther north, causing the current to extend excessively into the northern East/Japan Sea. This unrealistic northward intrusion resulted in a stronger and more extensive East Korea Warm

Current, contributing to an exaggerated warm bias in the region (Figs. 2f and 3f).

Both HYBRID and ZSTAR underestimated current speeds, as reflected in their bias and RMSE values. HYBRID exhibited a bias of -2.53 cm/s and an RMSE of 6.62 cm/s, while ZSTAR showed a slightly larger bias of -3.54 cm/s and an RMSE of 7.16 cm/s. The correlation with respect to GLORYS12 was high for both configurations, with a correlation coefficient of 0.84 and 0.83, indicating good overall agreement in capturing large-scale current patterns.


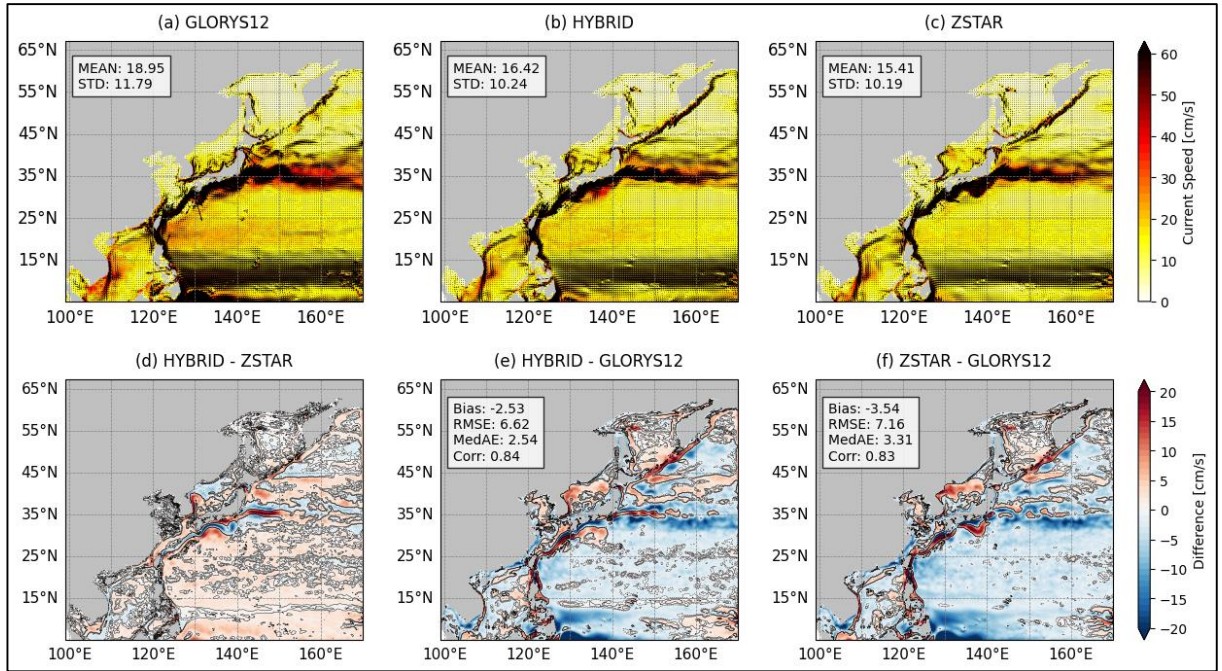

**Figure 9. Mean surface current speed from GLORYS12, HYBRID, and ZSTAR simulations. Black arrows indicate flow direction, with their size proportional to speed magnitude. (a–c) Spatial distributions of surface current speed with corresponding means and**





**standard deviations (STD). (d) Differences between HYBRID and ZSTAR. (e, f) Biases relative to GLORYS12, including mean**
**bias, RMSE, MedAE, and Corr. Contour lines in (d–f) indicate surface current speed biases ranging from -1.0 to 1.0**
**cm/s intervals.**

A comparison of the EKE between the HYBRID and ZSTAR configurations and GLORYS12 showed that both
configurations exhibited high spatial correlation with GLORYS12 (0.84), successfully capturing the general EKE
distribution (Fig. 10). However, both models tended to underestimate EKE across most regions, with mean biases of -51.95
cm²/s² for HYBRID and -60.12 cm²/s² for ZSTAR. Specifically, both configurations underestimated EKE in the southern
boundary regions, the Subtropical Counter Current region, and the Kuroshio and its extension, while overestimating EKE in
the Kuroshio recirculation region and the East/Japan Sea. ZSTAR underestimated EKE in the open ocean of the NWP but
overestimated it in the Kuroshio recirculation region and the East/Japan Sea, consistent with its higher bias in these areas. In
contrast, HYBRID exhibited a more significant underestimation of EKE in the Kuroshio and its extension region compared
to ZSTAR. HYBRID achieved a slightly lower RMSE (138.46 cm²/s²) than ZSTAR (138.86 cm²/s²). Additionally, the
MedAE was 30.37 cm²/s² for HYBRID and 39.71 cm²/s² for ZSTAR, suggesting that HYBRID performed slightly better in
capturing the magnitude of EKE variability across the domain.

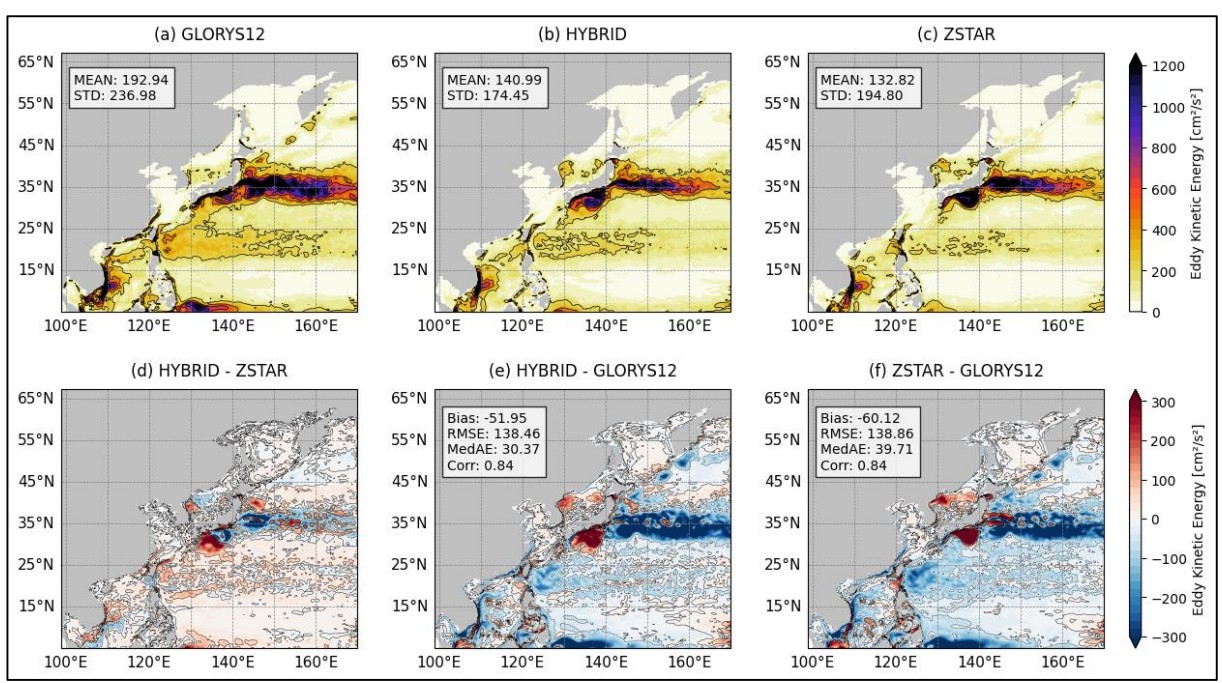


**Figure 10. Mean eddy kinetic energy (EKE) from GLORYS12, HYBRID, and ZSTAR simulations. (a–c) Spatial EKE distributions**
**with corresponding means and standard deviations (STD). (d) Differences between HYBRID and ZSTAR. (e, f) Biases relative to**
**GLORYS12, including mean bias, RMSE, MedAE, and Corr. Contour lines in (d–f) indicate EKE biases ranging from -0.5 to 1.0**
**cm²/s² at 0.5 cm²/s² intervals.**





### 3.3 Vertical structure and water masses

The vertical profiles of temperature and salinity biases for the HYBRID and ZSTAR configurations were compared against GLORYS12 and K-ORA22 across different regions in the NWP (Fig. 11). In the open ocean of the NWP (Fig. 11a), both configurations exhibited low biases and closely followed the vertical structures of temperature and salinity observed in GLORYS12. However, in the Kuroshio and its extension (Fig. 11b), the bias patterns differed between the two configurations. ZSTAR showed lower temperature and salinity biases up to 400 m compared to HYBRID, whereas HYBRID better maintained the vertical structure of GLORYS12 below 400 m, with temperature biases remaining below 0.3℃ and salinity biases under 0.05 psu.

In the Sea of Okhotsk (Fig. 11c), HYBRID exhibited larger negative biases than ZSTAR between 100 m and 600 m, with a temperature bias of approximately -1℃ at 400 m. The salinity bias for HYBRID was also more pronounced, reaching around -0.3 psu at 400 m, whereas ZSTAR showed relatively lower biases. In the East/Japan Sea (Fig. 11d), both models simulated salinity patterns similar to K-ORA22, but HYBRID consistently produced lower temperature biases, while ZSTAR exhibited a positive temperature bias of approximately 1℃ between 200 m and 400 m. In the Yellow Sea (Fig. 11e), both models displayed similar temperature and salinity profiles. The Yellow Sea is characterized by the Yellow Sea Bottom Cold Water Mass (YBCWM), a cold and dense water mass that forms near the bottom. However, both configurations showed positive temperature biases exceeding 2℃ near the bottom, suggesting limitations in accurately representing YBCWM. For salinity, both models exhibited large negative biases at the surface, with HYBRID and ZSTAR showing deviations of approximately -4 psu. This suggests that the river discharge forcing applied in both models overestimated freshwater input from major rivers in the region, such as the Yangtze and Yellow Rivers.



**Figure 11. Vertical mean profiles of temperature (left) and salinity (right) from GLORYS12 (green), K-ORA22 (orange), HYBRID (red), and ZSTAR (blue) across different Northwest Pacific (NWP) regions: (a) Open ocean area, (b) Kuroshio and its Extension, (c) Sea of Okhotsk, (d) East/Japan Sea, and (e) Yellow Sea. Biases for HYBRID (red dashed lines) and ZSTAR (blue dashed lines) are shown relative to the reference datasets (GLORYS12 or K-ORA22).**





**Figure 12. Meridional temperature section along 148°E from (a) GLORYS12 reanalysis, (b) HYBRID simulation, and (c) ZSTAR simulation, showing vertical temperature distribution. Panels (d), (e), and (f) illustrate temperature differences: HYBRID vs. ZSTAR (d), HYBRID vs. GLORYS12 (e), and ZSTAR vs. GLORYS12 (f). Contour lines in (d–f) indicate temperature biases ranging from -1.0 to 1.0 °C at 1.0 °C intervals.**

The NWP is characterized by two distinct water masses: Subtropical Mode Water (STMW) and North Pacific Intermediate Water (NPIW), both of which play crucial roles in the region's physical and biogeochemical processes. STMW is typically found within the upper thermocline at depths of 100–300 m, while NPIW occupies the intermediate layer, extending from approximately 300 to 800 m.

Figure 12 presents a comparison of the vertical temperature section along 148°E for HYBRID and ZSTAR against GLORYS12. Both configurations reproduced an overall temperature structure similar to that of GLORYS12, effectively





capturing the vertical thermal structure. However, ZSTAR exhibited a positive temperature bias exceeding 1℃ in high-latitude regions, while HYBRID showed a more pronounced positive bias of over 1.5℃ at depths where STMW is present.




**Figure 13. Meridional salinity section along 148˚E from (a) GLORYS12 reanalysis, (b) HYBRID simulation, and (c) ZSTAR simulation, showing vertical salinity distribution. Panels (d), (e), and (f) display salinity differences: HYBRID vs. ZSTAR (d), HYBRID vs. GLORYS12 (e), and ZSTAR vs. GLORYS12 (f). Red contour lines in (a–c) indicate σ₂ (density referenced to 2000 dbar) for each dataset. Contour lines in (d–f) represent salinity biases ranging from -0.1 to 0.1 psu at 0.1 psu intervals.**


Figure 13 compares the vertical salinity section along 148℃E between HYBRID, ZSTAR, and GLORYS12. HYBRID closely reproduced the NPIW salinity structure, while ZSTAR exhibited a salinity bias of approximately 0.3 psu, indicating challenges in accurately representing NPIW. When examining σ₂ (density referenced to 2000 dbar) within the range of 35.0



to 36.6, HYBRID accurately captured the thickness of the $\sigma_2$ layer associated with NPIW, whereas ZSTAR tended to

overestimate its thickness. However, both configurations showed a positive salinity bias of approximately 0.3 psu in STMW, suggesting a slight overestimation of salinity in this region.

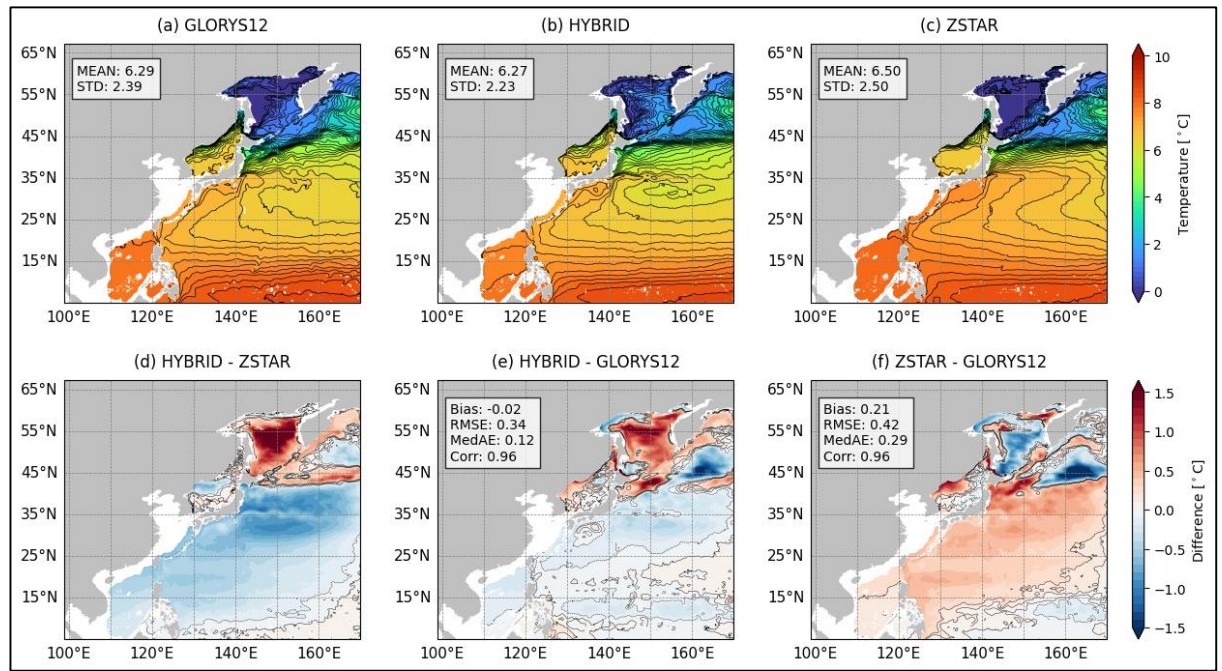

**Figure 14. Temperature distributions at depths corresponding to $\sigma_2$ = 35.8 from (a) GLORYS12, (b) HYBRID, and (c) ZSTAR**
**simulations, along with their respective means and standard deviations (STD). (d) Differences between HYBRID and ZSTAR. (e, f)**
**Biases relative to GLORYS12, including mean bias, root mean squared error (RMSE), median absolute error (MedAE), and**
**correlation (Corr). Contour lines in (d–f) indicate temperature biases ranging from -0.1 to 0.1 °C at 0.1 °C intervals.**

The σ2 value of 35.8 is defined as the salinity minimum layer of NPIW, and the depth at which this minimum layer is
located was extracted from each dataset. The temperature and salinity values at this depth from both configurations were then compared with those obtained from GLORYS12.

Figure 14 compares the temperature at the depth of the salinity minimum layer for both configurations against GLORYS12. HYBRID generally exhibited a spatial temperature distribution similar to GLORYS12, except for a positive temperature bias exceeding 1.0℃ in the OKH region. In contrast, ZSTAR tended to exhibit a positive bias across most regions, except for
areas influenced by open boundaries. Notably, ZSTAR exhibited a temperature bias of approximately 1℃ in the transition zone where the Kuroshio and Oyashio currents converge, a critical region for NPIW formation.

The salinity distribution at depths, where the salinity minimum layer is located was also compared between HYBRID and ZSTAR relative to the GLORYS12 (Fig. 15). In most regions, HYBRID showed minimal salinity biases relative to



GLORYS12, except for the OKH region, where it exhibited a positive salinity bias. In contrast, ZSTAR showed a salinity

bias of approximately 0.2 psu, except for areas influenced by open boundaries.

These results indicate that ZSTAR exhibited more significant positive temperature and salinity biases at depths where the minimum salinity layer is present compared to HYBRID. The larger biases observed in ZSTAR at intermediate depths are likely attributable to spurious diapycnal mixing, a well-known issue in ZSTAR configurations (Griffies et al., 2000). This excessive mixing can lead to artificial erosion of water mass properties, reducing the sharpness of the salinity minimum layer

and contributing to the observed biases.

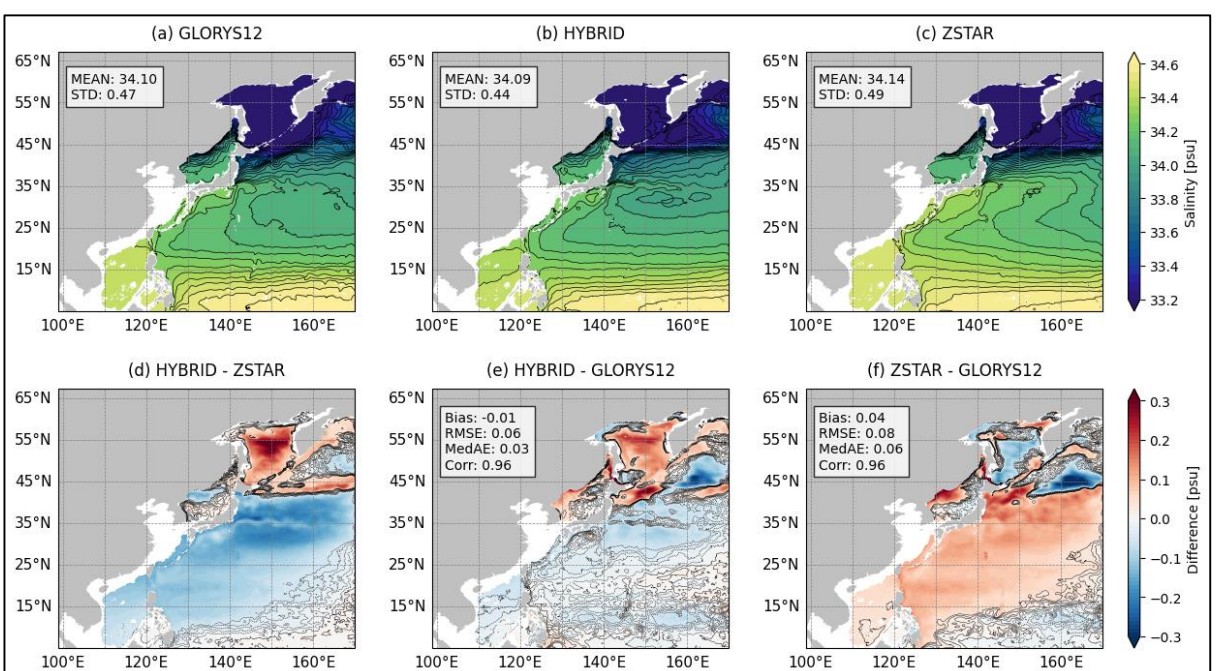

**Figure 15. Salinity distributions at depths corresponding to σ₂ = 35.8 from (a) GLORYS12, (b) HYBRID, and (c) ZSTAR simulations, along with their respective means and standard deviations (STD). (d) Differences between HYBRID and ZSTAR. (e, f)**

**Biases relative to GLORYS12, including mean bias, RMSE, MedAE, and Corr. Contour lines in (d–f) indicate salinity biases ranging from -0.1 to 0.1 psu at 0.1 psu intervals.**

To further investigate the differences in vertical structure and water mass representation between the HYBRID and ZSTAR configurations, an idealized age tracer experiment was conducted following the spin-up simulation. This experiment was

performed over a 10-year integration period to assess ventilation and subduction processes in both configurations. To facilitate comparison, the age tracer values were normalized following the approach of Adcroft et al. (2019). A value of zero represents older water that has remained in the interior for an extended period, while a value of 1 indicates younger water that has been more recently ventilated from the surface.



In the region where STMW is present, where HYBRID previously exhibited a positive temperature and salinity bias (Figs.
12e and 13e), the normalized age values were higher than those of ZSTAR (Fig. 16). This indicates that HYBRID simulates
younger water in this region, suggesting more active vertical ventilation or enhanced exchange with surface waters. This
over-ventilation may contribute to the observed positive temperature and salinity biases in STMW in HYBRID.

In high-latitude regions where ZSTAR exhibited positive temperature and salinity biases, the normalized age values were
lower in ZSTAR compared to HYBRID, indicating the presence of older water. At depths associated with NPIW formation,
ZSTAR showed higher normalized age values than HYBRID, meaning it simulated younger water in this critical layer.
These patterns suggest that spurious diapycnal mixing—a known issue in traditional Eulerian geopotential coordinate
models—plays a significant role in ZSTAR. In high-latitude regions, stronger diapycnal mixing in ZSTAR allows older
water from deeper layers to diffuse upward, leading to the observed positive temperature and salinity biases (Figs. 12f and
13f). Conversely, at NPIW depths, enhanced diapycnal mixing in ZSTAR facilitates the downward diffusion of young
surface water, disrupting the natural vertical separation of water masses and eroding the salinity minimum layer. While
HYBRID preserved vertical water mass properties more effectively, spurious diapycnal mixing in ZSTAR reduced its ability
to accurately simulate intermediate water properties, particularly in regions critical for NPIW formation (Figs. 13 and 15).

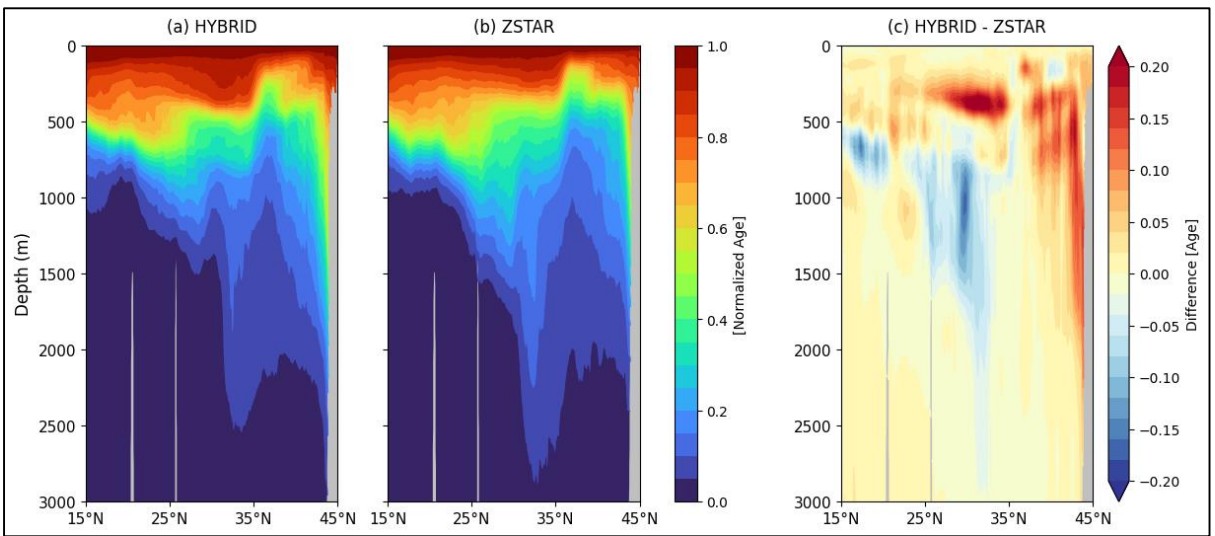

**Figure 16. Meridional normalized age tracer along 148°E for (a) HYBRID, (b) ZSTAR, and (c) their difference (HYBRID -**
**ZSTAR). The normalized age is computed as $f = (A_{max} - A)/A_{max}$, where $A_{max}$ is the maximum age in the simulation, following**
**Adcroft et al. (2019). Values range from 0 (oldest water) to 1 (youngest water), representing the relative ventilation age of water**
**masses.**

In the Yellow Sea, the YBCWM is a distinct water mass that plays a crucial role in shaping regional hydrography and
seasonal dynamics. It primarily forms in winter, when surface cooling induces vertical convection, allowing cold, dense





water to accumulate in the deeper regions of the Yellow Sea. As spring and summer progress, surface warming enhances stratification, effectively trapping the YBCWM at the subsurface. Accurate representation of the YBCWM is critical, as it influences regional circulation, water mass transformation, and broader atmospheric and biogeochemical processes. The
presence of the YBCWM modulates ocean-atmosphere interactions, potentially impacting typhoon intensification over the Yellow Sea basin (Moon and Kwon, 2012). Additionally, it plays a key role in regulating nutrient availability, oxygen dynamics, and primary production, thereby shaping regional biogeochemical cycling and ecosystem productivity (Huo et al., 2012; Su et al., 2013).

Figure 17 presents the bottom temperature distribution during summer and its bias relative to the K-ORA22 reanalysis. Since
GLORYS12 fails to represent the YBCWM entirely (Chang et al., 2024), K-ORA22 was used as the reference dataset for comparison. The YBCWM in summer is generally characterized by a circular water mass enclosed by the 10℃ isotherm near the bottom (Zhang et al., 2008; Li et al., 2021). This structure was well captured in K-ORA22, and both the HYBRID and ZSTAR configurations successfully reproduced its overall pattern.

However, as shown in Fig. 11e, both configurations exhibited a warm bias of approximately 2℃ at the bottom compared to
K-ORA22, with the bias being more pronounced in ZSTAR. Additionally, the YBCWM was shifted westward, resulting in a cold bias on the western side. Despite these biases, it is notable that both configurations successfully simulated the YBCWM without data assimilation—a significant improvement, as previous models have often failed to reproduce this feature due to the absence of explicit tidal forcing and reliance on parameterized tidal mixing.

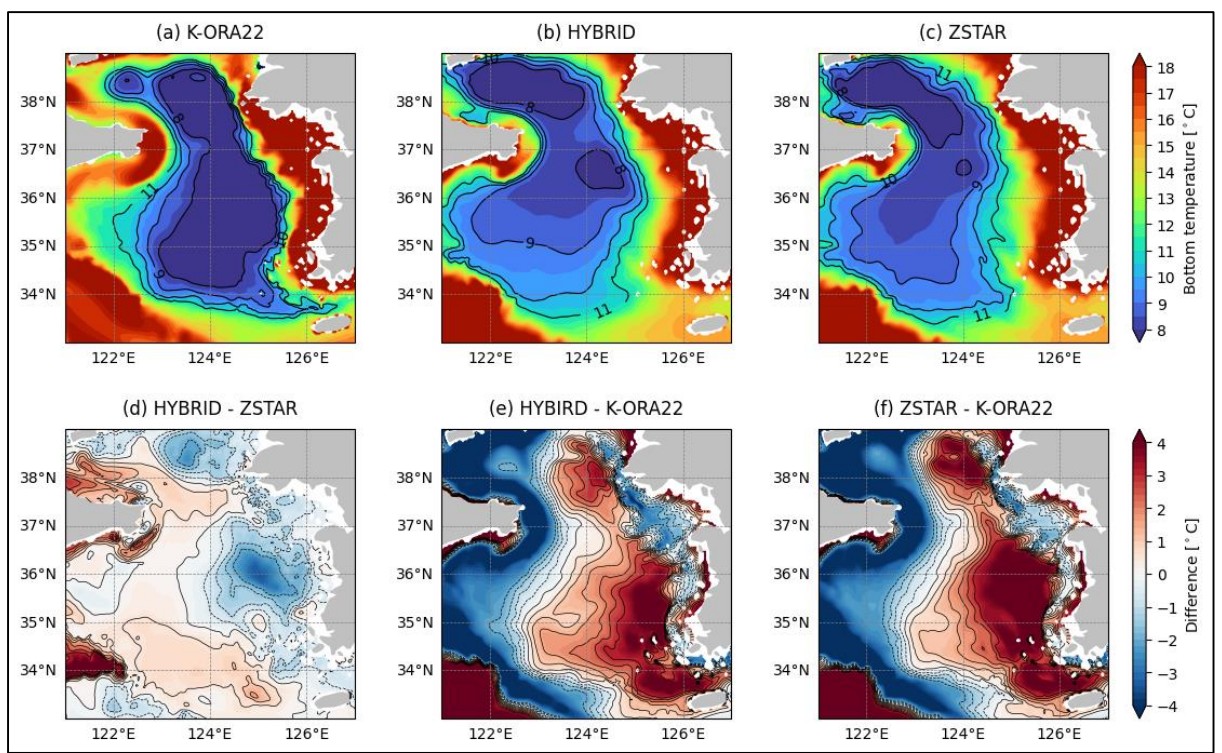





**Figure 17. Bottom temperature distributions in the Yellow Sea from (a) K-ORA22, (b) HYBRID, and (c) ZSTAR simulations. (d) Temperature difference between HYBRID and ZSTAR. (e, f) Biases relative to K-ORA22. Contour lines in (d–f) indicate temperature biases ranging from -2.0 to 2.0 °C at 0.5 °C intervals.**

### 3.4 Volume transport

The volume transport through key straits in the Northwest Pacific (NWP) was analyzed to evaluate the performance of the

HYBRID and ZSTAR configurations (Fig. 18). The Tokara Strait serves as a crucial passage for the Kuroshio Current, making it a key indicator of the current's transport dynamics. Meanwhile, the Korea/Tsushima Strait, Tsugaru Strait, and Soya Strait play significant roles in regulating the inflow and outflow of heat and salt from lower latitudes into the East Sea (ES) and their subsequent exchange with the open ocean. Given the limited availability of direct observational data for these straits, observed climatology or long-term mean values from previous studies were used for comparison.

In the Tokara Strait, the observed annual mean volume transport from 1987 to 2010 was 21.39 Sv (Wei et al., 2013). Both configurations overestimated this transport, with HYBRID simulating an annual mean of 28.40 Sv (overestimating by 7.01 Sv) and ZSTAR simulating 27.15 Sv (overestimating by 5.76 Sv). The transport magnitude was consistently higher in HYBRID than in ZSTAR.

For the Korea/Tsushima Strait, the observed annual mean transport, derived from sea-level differences (Shin et al., 2022),

was 2.61 Sv. HYBRID simulated a higher transport of 2.90 Sv, exceeding the observed value by 0.29 Sv, while ZSTAR underestimated it with an annual mean of 2.48 Sv, showing a negative bias of 0.13 Sv. Although HYBRID better captured the seasonal peak in transport during summer, its overestimation suggested an excessive flow through the strait.

According to acoustic Doppler current profiler measurements from 2003 to 2007, the mean volume transport through the Tsugaru Strait was 1.47 Sv (Han et al., 2016), consistent with previous estimates of approximately 1.5 Sv (Na et al., 2009;

Ohshima and Kuga, 2023). Both configurations overestimated this value, with HYBRID predicting an annual mean transport of 2.47 Sv (exceeding the observed value by 1.00 Sv) and ZSTAR predicting 2.04 Sv (overestimating by 0.57 Sv).

In the Soya Strait, the observed annual mean transport, estimated from high-frequency radar observations between 2003 and 2015 (Ohshima and Kuga, 2023), was 0.90 Sv. Both configurations underestimated the transport, with HYBRID simulating 0.57 Sv (0.33 Sv lower than observed) and ZSTAR showing 0.56 Sv (0.34 Sv lower).

Overall, the HYBRID configuration tended to overestimate transport, particularly through the Tokara and Tsugaru Straits, while underestimating it in the Soya Strait. In contrast, ZSTAR generally underestimated transport, as observed in the Korea/Tsushima Strait  and Soya Straits.



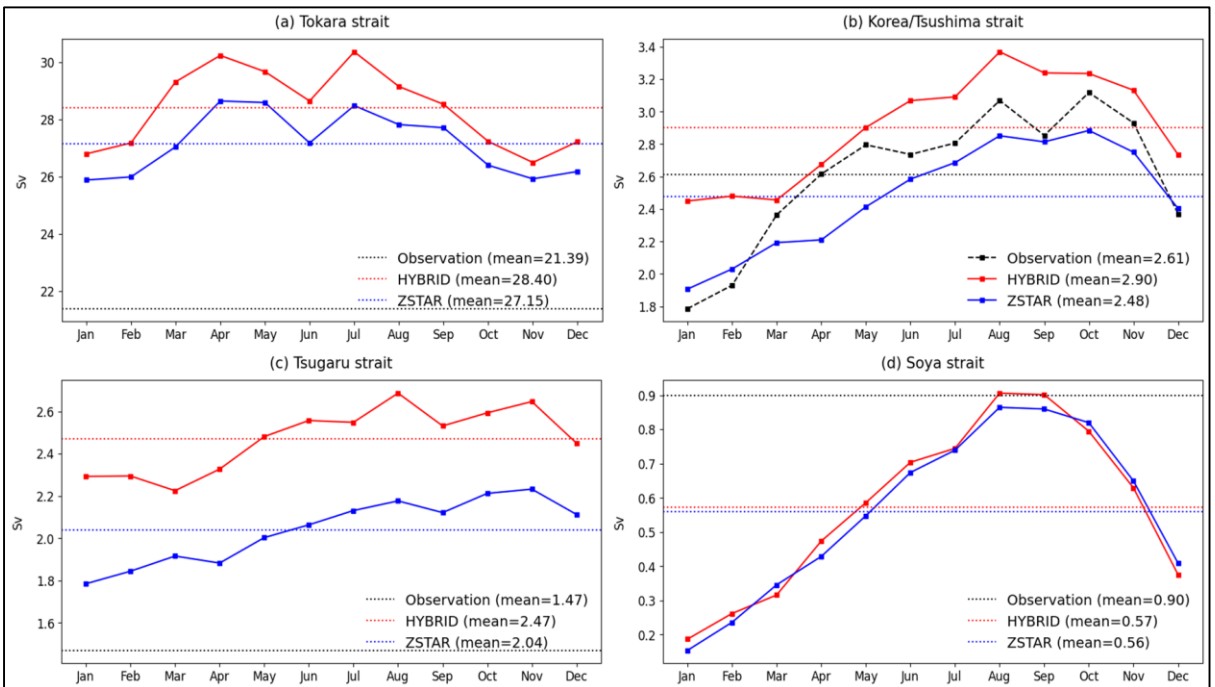

**Figure 18. Monthly climatological mean volume transport at (a) Tokara Strait, (b) Korea/Tsushima Strait, (c) Tsugaru Strait, and (d) Soya Strait. Observations (black) are compared with HYBRID (red) and ZSTAR (blue). Dotted lines represent the annual mean for each dataset, while solid lines show the monthly climatological mean.**

### 3.5 Tide simulation

Tides play a crucial role in shaping ocean dynamics in the NWP, where tidal forces strongly influence circulation, mixing, and water mass distribution. This is particularly evident in the Yellow Sea, which is characterized by exceptionally large tidal amplitudes, with a tidal range exceeding 8 m. Given the significant impact of tides on the physical and biogeochemical characteristics of this region, it is essential to assess the performance of tidal representations in these configurations.



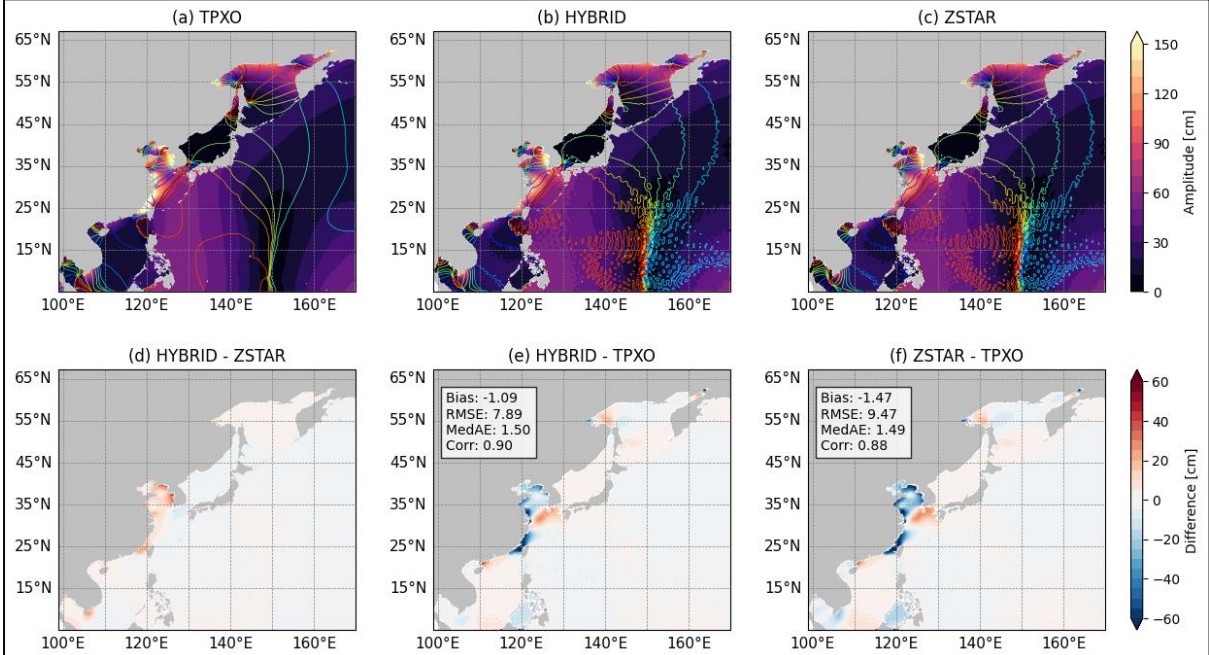

**Figure 19. Semidiurnal M2 tidal amplitude and phase from TPXO data, HYBRID, and ZSTAR simulations. Shaded contours represent tidal amplitude, while overlaid coloured contours show tidal phase for M2 (a–c). Panels below display tidal amplitude differences: (a) HYBRID vs. ZSTAR, (b) HYBRID vs. TPXO, and (c) ZSTAR vs. TPXO. Metrics include mean bias, RMSE, MedAE, and Corr.**

Figure 19 compares the tidal amplitude and phase of the semidiurnal M2 component between the HYBRID and ZSTAR configurations using TPXO, which served as the tidal boundary forcing dataset. Both configurations accurately simulated the tidal amplitude and phase, with HYBRID achieving a spatial correlation of 0.90 and ZSTAR showing 0.88, indicating a strong representation of tidal characteristics in the region. However, both models underestimated the M2 amplitude along the southeastern coast of China and in the Yellow Sea, while overestimating it in the Korea/Tsushima Strait. Notably, ZSTAR exhibited a more pronounced underestimation of the M2 amplitude than HYBRID in these regions.

Both configurations effectively simulated the K1 tidal amplitude and phase, with a high spatial correlation of 0.94 against TPXO data (Fig. S3). However, both models underestimated the K1 amplitude in the Yellow Sea, with a stronger bias in ZSTAR. In the sea of Okhotsk coastal region, HYBRID overestimated the amplitude, whereas ZSTAR showed a mixed bias, overestimating in the west and underestimating in the east, reflecting regional differences in tidal representation.

Overall, both configurations performed well in reproducing the amplitude and phase of the semidiurnal and diurnal tidal components. Nevertheless, the consistent underestimation of the Yellow Sea tidal amplitude across both configurations highlighted a common limitation. Importantly, the results suggested that tidal representation was influenced by the vertical coordinate system, with HYBRID showing better agreement with TPXO tidal amplitudes than ZSTAR. This suggests that



HYBRID may offer advantages in improving tidal simulations, particularly in regions with complex bathymetry and strong
tidal forcing.

**4 Discussion and conclusion**

In this study, we updated the base model of KOOS-OPEM, which had been developed using previous versions of the MOM series, to MOM6 to enhance regional ocean modeling capabilities. MOM6 introduced significant improvements in computational efficiency, numerical stability, and flexibility in vertical coordinate selection, enabling a more advanced
representation of oceanic processes. Given the increasing demand for accurate ocean predictions in the NWP and its marginal seas under a changing climate, this update aimed to improve the model's ability to represent key oceanic physical dynamics, current systems, and the physical characteristics of major marginal seas. Comprehensive sensitivity experiments were conducted to evaluate performance differences between the ZSTAR coordinate system, used in previous models, and the HYBRID system within MOM6's Lagrangian remapping framework. To ensure a robust assessment, both configurations
were compared against multiple observational datasets and two reanalysis products, GLORYS12 and K-ORA22, providing insights into how vertical coordinate systems influence the reproduction of key physical and dynamical features of the NWP. The results revealed significant differences between the HYBRID and ZSTAR configurations while also highlighting shared limitations in representing certain oceanographic variabilities.

A comparison of modeled SST with OISST satellite observations showed that both configurations effectively captured
seasonal SST patterns and gradients, demonstrating strong agreement with the OISST dataset (Fig. 2 and Fig. 3). However, both exhibited warm biases during winter, particularly in the Kuroshio and its extension and East/Japan sea. In Kuroshio-Oyashio Transition Zone, HYBRID showed a more pronounced warm bias compared to ZSTAR, whereas in East/Japan sea, ZSTAR exhibited a larger warm bias. The exaggerated warm bias in the East/Japan Sea for ZSTAR was closely linked to differences in representing the East Korea Warm Current, a critical feature influencing regional temperature distribution (Fig.
9). Both configurations overestimated East Korea Warm Current strength, leading to an overshoot in the current. However, HYBRID exhibited only a slight overshoot and captured the separation point more accurately than ZSTAR. In contrast, ZSTAR simulated a northward displacement of the separation point, causing excessive extension of the East Korea Warm Current, which intensified the warm bias in northern East/Japan Sea (Figs. 2f and 3f).

The evaluation of vertical temperature and salinity structures provided further insights into differences between HYBRID
and ZSTAR. Across most regions, both configurations successfully reproduced vertical hydrographic properties comparable to those in reanalysis datasets. However, notable discrepancies emerged in their representation of specific water masses.

The NPIW was represented more accurately in HYBRID than in ZSTAR. HYBRID closely captured the thickness and vertical structure of the isopycnal layer associated with NPIW (Fig. 13) and exhibited lower salinity biases compared to GLORYS12. In contrast, ZSTAR overestimated the thickness of the $\sigma_2$ layer associated with NPIW and showed a salinity




bias of approximately 0.2 psu. These differences were attributed to spurious diapycnal mixing inherent in the traditional ZSTAR system, which disrupted stratification and reduced the accuracy of intermediate water properties.

The idealized age tracer experiment further clarified these discrepancies (Fig. 16). At depths associated with NPIW formation, ZSTAR exhibited higher normalized age values than HYBRID, indicating the simulation of younger water masses in these layers. This suggested that enhanced diapycnal mixing in ZSTAR facilitated downward diffusion of younger

surface waters, eroding the salinity minimum layer that defines NPIW. In contrast, HYBRID preserved vertical stratification, leading to a more accurate representation of NPIW. However, in the STMW region, HYBRID displayed positive temperature and salinity biases due to excessive ventilation. The normalized age tracer results indicated that HYBRID simulated younger water in the STMW, with higher normalized age values than ZSTAR. This suggested that HYBRID promoted more active vertical ventilation or enhanced surface water exchange. While such processes facilitate surface-to-

interior interactions, they likely contributed to the observed temperature and salinity biases (Figs. 12 and 13).

However, HYBRID exhibited poorer performance than ZSTAR in high-latitude regions, as indicated by larger temperature and salinity biases between depths of 100 m and 600 m (Fig. 11c). This discrepancy was primarily due to a common limitation of isopycnal coordinates: poor vertical resolution in weakly stratified regions, which are characteristic of high latitudes (Adcroft et al., 2019). A comparison of active layers between HYBRID and ZSTAR (Fig. S4) revealed that

HYBRID generally maintained fewer active layers, particularly in weakly stratified regions. This reduction in active layers likely contributed to the increased temperature and salinity biases observed in HYBRID, underscoring the challenges of using isopycnal coordinates in high-latitude environments.

To improve water property representation in these regions, adjustments to the maximum layer thickness or modifications to the target density profile could enhance vertical resolution and better capture stratification. Such refinements may mitigate

resolution loss and reduce temperature and salinity biases in HYBRID.

Both configurations successfully reproduced the overall structure of the YBCWM despite the absence of data assimilation (Fig. 17). However, notable differences were observed, with HYBRID demonstrating a more accurate representation of temperature structure than ZSTAR. This improvement was closely linked to HYBRID's better representation of seasonal stratification. A well-defined seasonal stratification is crucial for YBCWM formation, as it reduces excessive vertical mixing

in summer and allows cold water to persist near the bottom. Given the sensitivity of YBCWM formation to vertical mixing, understanding the mechanisms governing stratification is essential for improving its representation in ocean models.

To further investigate the processes influencing its formation, sensitivity experiments conducted with and without the shear-driven mixing parameterization (Jackson et al., 2008) revealed that this parameterization played a crucial role in shaping and maintaining the YBCWM (not shown here). However, despite its effectiveness in reproducing the YBCWM structure, the

shear-driven mixing parameterization (Jackson et al., 2008) tended to induce excessive mixing in certain shelf regions with strong tidal forcing (Drenkard et al., submitted). Therefore, future work should focus on optimizing the turbulent decay length scale in the Jackson parameterization to better regulate mixing intensity in these regions (Drenkard et al., submitted).



Building on these findings, the evaluation of tidal dynamics further highlighted differences between the HYBRID and ZSTAR configurations. Both effectively simulated the semidiurnal (M2) and diurnal (K1) tidal amplitudes and phases across the NWP, demonstrating their ability to reproduce key tidal characteristics. However, HYBRID outperformed ZSTAR in capturing the barotropic tidal amplitude in the Yellow Sea, particularly for the M2 tide. Several studies have emphasized that barotropic tides in this region are seasonally modulated by stratification through its influence on bottom friction and energy dissipation (e.g., Kang et al., 2002; Müller et al., 2014). A comparison of the YBCWM further supports the differences in stratification representation between the two configurations. HYBRID exhibited a lower temperature bias near the bottom compared to ZSTAR, suggesting that it better captured seasonal stratification in the Yellow Sea. Since seasonal stratification directly influences both YBCWM formation and internal tide modulation, HYBRID's improved tidal amplitude simulation is likely linked to its enhanced representation of stratification. Given the strong dependence of baroclinic tides on stratification and vertical mixing, the choice of vertical coordinate system plays a crucial role in accurately capturing these processes. While HYBRID demonstrated improved tidal amplitude reproduction in the Yellow Sea, further investigation is needed to clarify the mechanisms through which different vertical coordinates influence tidal dynamics, particularly the generation, propagation, and dissipation of baroclinic tides.

Both HYBRID and ZSTAR struggled to accurately represent sea surface salinity, particularly in areas affected by river discharge (Fig. 4 and Fig. 5). Despite applying bias correction to GLOFAS, both models overestimated the freshwater influence, leading to significant negative salinity biases, exceeding -1.0 psu, in the Yellow Sea and East China Sea. To address this issue, repositioning the Yangtze River mouth further inland may better capture its interactions with the coastal ocean. Additionally, further investigation into other major rivers, such as the Yellow River, and additional bias corrections are essential to improve freshwater dynamics representation in the region.

Both configurations effectively captured the overall spatial distribution of SSH in the NWP, demonstrating strong agreement with observed large-scale patterns (Fig. 6). However, when SSH variability was analyzed separately into large-scale and mesoscale components using a two-year cut-off period with high- and low-pass filters, both models significantly underestimated variability magnitude compared to observations. For large-scale variability (Fig. 7), the models failed to fully capture SSH variability in dynamic regions such as the Kuroshio and its extension and the North Equatorial Current. Similarly, mesoscale variability, influenced by eddies and smaller-scale processes, was also underestimated, with both configurations showing reduced intensity and weaker high-frequency fluctuations (Fig. 8). This underestimation extended to the EKE, where both HYBRID and ZSTAR failed to reproduce the observed magnitude, particularly in regions of strong mesoscale activity, such as the K-KE. While the models replicated spatial patterns and variability correlations, their inability to resolve the intensity of large- and mesoscale dynamics underscores a key limitation in accurately simulating the energetic processes defining the NWP. Addressing these limitations requires sensitivity experiments on horizontal viscosity, which plays a crucial role in modulating mesoscale and sub-mesoscale dynamics in ocean models. Excessive viscosity can overly dampen eddy activity and high-frequency fluctuations, leading to an underestimation of SSH variability and EKE, as observed in both configurations. Conversely, insufficient viscosity may introduce numerical instabilities, particularly in



strong current regions such as the Kuroshio and its extension. Optimizing viscosity parameters through targeted sensitivity experiments can help balance numerical stability and realistic energy dissipation, ultimately improving the model's ability to resolve large- and mesoscale variability in the NWP.

In summary, the HYBRID configuration demonstrated notable advantages over ZSTAR in several key aspects of NWP simulation. It effectively captured stratification, reduced spurious diapycnal mixing, and provided more accurate representations of features such as the NPIW, tidal dynamics in the Yellow Sea, and the East Korea Warm Current separation. These improvements align with findings from Adcroft et al. (2019), who showed that ZSTAR induces significant warm drift in intermediate layers due to excessive diapycnal mixing, whereas HYBRID mitigates this issue by better
preserving water mass properties. Given that HYBRID has proven effective not only in global ocean simulations but also in regional modeling experiments, it shows promise as a robust vertical coordinate system for high-resolution regional applications, particularly in strongly stratified environments such as the NWP. However, HYBRID also exhibited limitations, particularly in high-latitude regions where weak stratification led to significant vertical structure biases and in the STMW region, where overly active ventilation introduced temperature and salinity biases. To overcome these limitations and further
optimize the HYBRID configuration, refinements in vertical resolution are necessary, particularly in weakly stratified high-latitude regions and STMW formation zones. Adjusting maximum layer thickness in these areas could help mitigate vertical resolution loss and reduce temperature and salinity biases. Additionally, refining target density profiles to better capture regional stratification characteristics may enhance the model's ability to represent key water mass properties more accurately. Future work should also explore the impact of horizontal viscosity tuning to improve mesoscale energy representation and
enhance eddy-driven process simulation. By addressing these issues, the HYBRID coordinate system can be further refined to provide a more robust and accurate framework for high-resolution regional ocean modeling in the NWP.

Beyond improvements in physical ocean modeling, a coupled physical-biogeochemical model is critical for a comprehensive understanding of ecosystem dynamics in the NWP. The NWP contains several ecologically significant regions, including the East/Japan Sea and Kuroshio-Oyashio Transition Zone, which support diverse marine ecosystems and essential fisheries. To
fully capture these ecosystems' complexity, biogeochemical models must be integrated with physical models, allowing for a more detailed understanding of nutrient cycling, primary productivity, and ecosystem responses to environmental changes, particularly in the face of shifting climatic conditions and increasing anthropogenic pressures. Therefore, coupling COBALT with KOOS-OPEM, based on MOM6, is expected to provide a comprehensive tool for simulating both physical and biogeochemical dynamics in the NWP. This coupled system will enable more accurate predictions of key biogeochemical
processes, using dynamic downscaling to assess their responses to environmental changes and long-term oceanic trends. Such efforts are crucial for advancing sustainable resource management and ensuring the long-term resilience of marine ecosystems in the NWP.



**Code availability**

The source code for each model component has been archived at Zenodo (Chang et al., 2025c; https://doi.org/10.5281/zenodo.15054440). The MOM6 code is available on GitHub at mom-ocean/MOM6 and NOAA-GFDL/MOM6. Additional repositories for other model components can be found at NOAA-GFDL's GitHub. Scripts for generating regional MOM6 initial and boundary conditions, along with other required inputs and diagnostics, are maintained in the NOAA CEFI GitHub repository: https://github.com/NOAA-GFDL/CEFI-regional-MOM6/.


**Data availability**

All model output used in this study is available at Zenodo (Chang et al., 2025a; https://doi.org/10.5281/zenodo.15054536). The corresponding model parameters, forcing data, and initial condition files have been archived at Zenodo (Chang et al., 2025b; https://doi.org/10.5281/zenodo.15054924). The datasets used for model validation and comparison are summarized in

Table 2, along with their respective URLs or DOIs for access. These include OISSTv2.1 (NOAA NCEI, Huang et al., 2021), GLORYS1212 reanalysis (DOI: 10.48670/moi-00021, Lellouche et al., 2021), K-ORA22 reanalysis (DOI: 10.1016/j.pocean.2024.103359, Chang et al., 2024); de Boyer Montégut's global ocean mixed layer depth dataset (DOI: 10.17882/98226, de Boyer Montégut, 2024), the Global Ocean Gridded L4 Sea Surface Heights and Derived Variables dataset (DOI: 10.48670/moi-00148; CMEMS, 2023), and the OSU TPXO9 Tide Model (TPXO9, Egbert and Erofeeva,

800    2002).

**Author contribution**

Conceptualization: IC, YHK, Y-GP, and RH. Model configuration: IC, YHK, Y-GP, HJ, GP, ACR, and RH. Model simulations: IC. Model evaluation: IC, YHK, and Y-GP. Formal analysis: IC, YHK, and Y-GP. Visualization: IC and YHK.

Original draft: IC and YHK. Review and editing: IC, YHK, Y-GP, HJ, GP, ACR, and RH.

**Competing interests**

The authors declare that they have no conflict of interest

**Acknowledgements**

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
