# Peer review of "Assessing Vertical Coordinate System Performance in the Regional Modular Ocean Model 6 configuration for Northwest Pacific"

_EGUsphere, 2025_

## Referee Comment (RC1)

Review of

**"Assessing Vertical Coordinate System Performance in the Regional Modular Ocean Model 6 configuration for Northwest Pacific" (https://doi.org/10.5194/egusphere-2025-3211)**

by I. Chang et al.

22nd August 2025

**General comments**

This manuscript aims at investigating the sensitivity of a regional MOM6-based mesoscale-rich ocean configuration of the Northwest Pacific (NWP) to the type of vertical coordinate system employed. Two configurations differing only on the vertical grid are compared: one, named ZSTAR, adopts a widely used quasi-Eulerian z*-scheme while the second, named HYBRID, employs a flexible hybrid z*-isopycnal Arbitrary Lagrangian-Eulerian vertical coordinate.

The assessment is conducted comparing model simulations against observational and reanalysis data for the surface and vertical structure of the active tracers, MLD, SSH, surface currents and EKE, volume transports and tidal amplitudes and phases. The methodology is quite straightforward, and the analysis includes standard statistics.

Results indicate that at the surface, both configurations present similar biases while at depth the HYBRID configuration is able to reduce spurious diapycnal mixing in the interior of the deep ocean, significantly improving the representation of the North Pacific Intermediate Water in comparison to the ZSTAR configuration. HYBRID shows superior skill over ZSTAR also in simulating tides in shallow regions such as the Yellow Sea. However, HYBRID presents significant biases in representing upper and mid-depth water masses at high latitudes, a problem that the authors attributes to the coarse vertical resolution that the HYBRID configuration tends to have in weakly stratified regions. The latter is also a well-known limitation of classical isopycnal models.

The analysis is solid and the study quite innovative and informative, being isopycnal coordinates (either pure or hybridised) rarely used in regional configurations where the stratification can be quite weak, especially in shallow areas with strong tides. The manuscript is very well written and fits well the scope of the journal. However, I do have few remarks/questions that might require some additional simulations. For this reason, publication is recommended after major revision.

**Major remarks**

My main criticism concerns the setup of the vertical grid of the HYBRID configuration. As stated by the authors, HYBRID uses the same target densities of Adcroft et al. 2019, a global ocean configuration where the target density range was chosen to fulfil the needs of a model domain covering the entire globe. In the case of a regional model like the one described in this paper, the extension of the model domain is much more limited in the meridional direction, allowing one to use a narrower range of target densities. I think that Fig. S4 and the biases that HYBRID presents at the high-latitudes clearly demonstrate that the chosen target density range was not optimal for the domain of this study – according to Fig. S4, HYBRID seems to have on average ~ 10 active layers less than ZSTAR! Consequently, I believe the comparison between the two vertical coordinate system is not fair, and I think it would be great if the authors could add one more simulation where the target density range is more tailored to the regional domain of this study. This additional simulation could allow us to understand whether the problems that the authors are reporting in the case of HYBRIDS are due to its not-optimal setup or there are intrinsic limitations in the z*-isopyncal vertical coordinate.

**Specific comments**

**L89**: MOM6 introduces a "significantly different algorithm" compared to previous versions -> can you be a bit more specific?

**L149**: "layer thickness of 2 meters extending to a depth of 14 meters in the ZSTAR space." -> does it mean that the thickness of the upper layer is 2m? If yes, I think it could be too thick – e.g. Bernie et al 2005 and Siddorn & Furner 2012 recommend a resolution of ~1m to properly model diurnal SST variability – and this could possibly be one of the causes behind the SST biases you are seeing in both ZSTAR and HYBRID, please clarify.

**L151-155**: "The transition depth between the isopycnal and ZSTAR coordinates deepened toward higher latitudes (Adcroft et al., 2019)" -> could you please clarify how the hybrid vertical coordinate works – a sketch or plot showing a vertical section with grid layers and the topography would be very helpful.

Also, it is not clear to me what happens on the shelf: because of the shallow depths, the unstratified bottom boundary layer can be quite thick, and is not unusual that it merges with the upper mixed layer, generating a well mixed water column, especially in regions where tides are strong. In such a regime, I see the usage of isopycnal coordinates quite challenging ... for example, according to Fig. S4 it seems that in shallow areas HYBRID has much less active layers than ZSTAR ...

**L148-157**: what formulation of bottom friction is used by the two configurations? What type of lateral boundary conditions are used in the case of the ZSTAR model? How is represented the bottom topography with ZSTAR, with full or partial steps?

**L251**: "where u' and v' represent deviations of the zonal and meridional velocity components from their respective means." -> averages over which period?

**L322-324**: It seems to me that in the South China Sea and across the open ocean it is actually the contrary - i.e., HYBRID has a slightly larger positive bias than ZSTAR - see Fig 4 – please clarify.

**L336**: ", with ZSTAR showing stronger positive biases in the open ocean and ..." -> It seems tom me that in the open ocean the two model are in the same ballpark, with ZSTAR sightly better than HYBRID in terms of warm bias ... please clarify.

**L341-344**: you already said this at L336-337 - you may want to merge and simplify the text here.

**L356**: "15 m compared to ZSTAR in the open ocean" -> perhaps "Southern open ocean"?

**L362**: Perhaps you were meaning Fig. S2?

**Fig. 9** -> Vectors are not clear (I can not see them), perhaps you may consider avoiding plotting them ...

**L456:** "HYBRID achieved a slightly lower RMSE (138.46 cm$^2$/s$^2$) than ZSTAR (138.86 cm$^2$/s$^2$)" -> to me, they are very similar ...

**L478**: "The Yellow Sea is characterized by the Yellow Sea Bottom Cold Water Mass (YBCWM), a cold and dense water mass that forms near the bottom. However, both configurations showed positive temperature biases exceeding 2°C near the bottom, suggesting limitations in accurately representing YBCWM" -> is this an indication that actually in the Yellow Sea the two models are using a similar vertical discretization – mainly z* levels? Fig. S4 seems to support this conclusion (the difference in active layers in the Yellow Sea is <= 6). This links to the previous comment: it would be great if you could clarify how the hybrid vertical coordinate works in shallow areas (a plot would be very helpful).

**Fig. 11**:  Are the discontinuities in the HYBRID profiles seen in, e.g., panel (b) at around 800m or panel (c) around 400m, due to the transition from z* to isopycnal levels?

**L620-622**: "Overall, the HYBRID configuration tended to overestimate transport, particularly through the Tokara and Tsugaru Straits, while underestimating it in the Soya Strait. In contrast, ZSTAR generally underestimated transport, as observed in the Korea/Tsushima Strait and Soya Straits." Could the authors try to explain possible reasons for this? Could it be due to the fact that both configurations use the same formulation and coefficients for the bottom friction, but the ZSTAR configuration has some additional drag from the lateral boundary conditions (assuming ZSTAR uses a no-slip LBC, please clarify)?

**L652-655**: This seems to be in agreement with Wise et al. 2022, that shows that better resolving the bottom topography (terrain following and multi-envelope vertical coordinates versus z*-levels with partial steps) allows for a more accurate

representation of the tides. Also, Graham et al. 2018 and Bruciaferri et al. 2022 showed the detrimental impact of the 10m minimum depth approximation on the tidal propagation, especially in shallow areas. This could explain the consistent underestimation of tidal amplitude in both configurations in the Yellow Sea. Please consider discussing here how the two configurations differ in the representation of the terrain and how this could explain the better performance of HYBRID.

**L659:** "computational efficiency"-> can the authors quantify this for this domain?

**L683:** "lower salinity biases" -> but the temperature biases seem to me much larger (Fig 12e) – please clarify.

**L694-695**: "While such processes facilitate surface-to-interior interactions, they likely contributed to the observed temperature and salinity biases (Figs. 12 and 13)." -> could the authors please clarify a possible mechanism for this?

**L701-702**: "This reduction in active layers likely contributed to the increased temperature and salinity biases observed in HYBRID, underscoring the challenges of using isopycnal coordinates in high-latitude environments." -> I think Fig S4 shows that HYBRID has fewer active layers not just at high latitudes, but everywhere .... As I mentioned before, are we sure these are intrinsic difficulties of the hybrid z*-isopycnal vertical coordinate and not due to the not-optimal configuration of the HYBRID vertical grid?

**L703-704:** "modifications to the target density profile could enhance vertical resolution and better capture stratification" -> could the authors clarify why they chose to use a global setup in a regional model?

**L725**: "HYBRID's improved tidal amplitude simulation is likely linked to its enhanced representation of stratification." -> or to a better representation of flow-topography interactions in comparison to ZSTAR?

**L729-731**: "further investigation is needed to clarify the mechanisms through which different vertical coordinates influence tidal dynamics, particularly the generation, propagation, and dissipation of baroclinic tides." -> The 10m minimum depth approximation could be one possible candidate (see Graham et al. 2018 and Bruciaferri et al. 2022).

References

D.J. Bernie, S.J. Woolnough, J.M. Slingo, E. Guilyardi: Modeling diurnal and intraseasonal variability of the ocean mixed layer Journal of Climate, 18 (2005), pp. 1190-1202, https://doi.org/10.1175/JCLI3319.1

Bruciaferri, D., Tonani, M., Ascione, I., Al Senafi, F., O'Dea, E., Hewitt, H. T., and Saulter, A.: GULF18, a high-resolution NEMO-based tidal ocean model of the Arabian/Persian

Gulf, Geosci. Model Dev., 15, 8705–8730, https://doi.org/10.5194/gmd-15-8705-2022, 2022.

Graham, J. A., O'Dea, E., Holt, J., Polton, J., Hewitt, H. T., Furner, R., Guihou, K., Brereton, A., Arnold, A., Wakelin, S., Castillo Sanchez, J. M., and Mayorga Adame, C. G.: AMM15: a new high-resolution NEMO configuration for operational simulation of the European north-west shelf, Geosci. Model Dev., 11, 681–696, https://doi.org/10.5194/gmd-11-681-2018, 2018.

Siddorn, J. R. and Furner, R.: An analytical stretching function that combines the best attributes of geopotential and terrain-following vertical coordinates, Ocean Model., 66, 1–13, https://doi.org/10.1016/j.ocemod.2013.02.001, 2013.

Wise, A., Harle, J., Bruciaferri, D., O'Dea, E., and Polton, J.: The effect of vertical coordinates on the accuracy of a shelf sea model, Ocean Model., 170, 101935, https://doi.org/10.1016/j.ocemod.2021.101935, 2021

---

## Author Comment (AC6)

[revised manuscript text omitted]

---

## Author Response (AR1)

**Author's General Response**

We sincerely thank the reviewer for their thoughtful evaluation and constructive comments. We greatly appreciate the reviewer's recognition of the scientific merit and clarity of our work, as well as the insightful suggestions that have helped us strengthen the manuscript.

[Figure]

**Figure S5. Spatial distribution of active layers in HYBRID and ZSTAR on December 22, 2012. (a, b) Number of active layers in HYBRID and ZSTAR, respectively, where an active layer is defined as having a thickness greater than 0.001 m. (c) Difference (HYBRID - ZSTAR).**

In particular, we fully agree with the reviewer's main remark regarding the use of a global target density range for the HYBRID configuration. Following this valuable suggestion, we conducted an additional simulation using a regional target density array tailored to the hydrographic characteristics of the Northwest Pacific domain. This modification significantly improved the vertical discretization in HYBRID, increasing the number of active layers (as shown in the revised Fig. S5). As a result, the revised HYBRID configuration better captured the vertical structure of the North Pacific Intermediate Water and reduced the positive temperature bias in the subtropical region. In particular, the regional target density setup enhanced the simulation of the East Sea Intermediate Water, which was previously too weak in the global-density configuration.

[Figure]

**Figure 1** Meridional salinity section along 131˚E from (a) GLORYS12 reanalysis, (b) HYBRID simulation, and (c) ZSTAR simulation, showing vertical salinity distribution.

We have incorporated these new results into the revised manuscript. Overall, the reviewer's comments have led to a more balanced and comprehensive assessment of the vertical coordinate sensitivity in the regional MOM6 framework.

The detailed responses to the reviewer's specific comments are provided below.

**L89: MOM6 introduces a "significantly different algorithm" compared to previous versions -> can you be a bit more specific?**

➔ Thank you for pointing this out. We have revised the sentence to clarify the specific algorithmic differences introduced in MOM6 compared to previous versions. In addition to the adoption of a vertically Lagrangian finite-volume framework, MOM6 transitions from a B-grid to a C-grid discretization, which improves the representation of momentum and tracer advection. Furthermore, the ocean boundary layer mixing parameterization has been updated from the traditional K-Profile Parameterization (Large et al. 1994) to the energetics based planetary boundary layer (Reichl and Hallberg, 2018) scheme, enhancing the simulation of vertical mixing and surface boundary processes. These algorithmic and physical updates collectively contribute to the improved numerical stability and physical realism of MOM6.

➔ **(Revised text)** MOM6 introduces a significantly different algorithm compared to previous versions (up to MOM5) and offers substantial improvements in computational efficiency

and stability. A key advancement is its use of vertical Lagrangian remapping (Griffies et al., 2020), a variant of the Arbitrary Lagrangian–Eulerian (ALE) algorithm, which allows for the implementation of various vertical coordinate systems, including geopotential (z or z*), isopycnal, terrain-following, or hybrid/user-defined coordinates. MOM6 also adopts a C-grid discretization instead of the previous B-grid and updates the ocean boundary layer parameterization from the traditional K-Profile Parameterization (Large et al. 1994) to the energetically consistent planetary boundary layer (ePBL) scheme (Reichl and Hallberg, 2018), further improving vertical mixing representation and surface–interior coupling. Additionally, newly developed open boundary conditions and improved regional modeling capabilities in MOM6 facilitate its effective use in regional ocean models. (line 84-90)

**L149: "layer thickness of 2 meters extending to a depth of 14 meters in the ZSTAR space." -> does it mean that the thickness of the upper layer is 2m? If yes, I think it could be too thick – e.g. Bernie et al 2005 and Siddorn & Furner 2012 recommend a resolution of ~1m to properly model diurnal SST variability – and this could possibly be one of the causes behind the SST biases you are seeing in both ZSTAR and HYBRID, please clarify.**

➔ In this study, the uppermost layer in the ZSTAR vertical grid configuration has a thickness of 2 m, extending to a depth of 14 m. While this design provides a reasonable balance between computational efficiency and representation of the mixed layer, it may be relatively coarse for capturing processes that operate at sub-daily scales. Previous studies (e.g., Bernie et al., 2005; Siddorn & Furner, 2012) have shown that vertical resolutions of ~1 m or finer are required in the surface ocean to adequately resolve diurnal warming and associated SST variability. Therefore, the 2 m upper layer thickness employed here may contribute to the persistent SST warm biases identified in both the ZSTAR and HYBRID simulations. We note that although our primary focus was on large-scale circulation and stratification, rather than diurnal processes, this limitation could influence the realism of the surface heat budget, particularly under strong insolation conditions. Future experiments employing enhanced vertical resolution in the upper ocean (e.g., 1 m thickness for the first few layers) will be necessary to further assess the extent to which layer thickness impacts the SST biases and to determine whether improving vertical discretization can reduce the discrepancy between model results and observations. We will clarify this point and discuss its possible link to the SST biases in the Discussion section.

➔ **(Revised text)** However, both exhibited warm biases during winter, particularly in the Kuroshio and its extension and East/Japan sea. These SST biases may partly stem from the relatively coarse vertical resolution near the surface, where the uppermost layer thickness

of 2 m in the ZSTAR grid (and similarly in HYBRID) can limit representation of diurnal SST variability (Bernie et al., 2005; Siddorn and Furner, 2012). Such resolution may underestimate sub-daily mixing and surface heat exchange, contributing to persistent warm biases under strong insolation conditions. Future sensitivity experiments with finer near-surface resolution (e.g., 1 m thickness for the upper layers) are planned to evaluate whether enhancing vertical discretization can mitigate these SST biases. (line 693-697)

**L151-155: "The transition depth between the isopycnal and ZSTAR coordinates deepened toward higher latitudes (Adcroft et al., 2019)" -> could you please clarify how the hybrid vertical coordinate works – a sketch or plot showing a vertical section with grid layers and the topography would be very helpful. Also, it is not clear to me what happens on the shelf: because of the shallow depths, the unstratified bottom boundary layer can be quite thick, and is not unusual that it merges with the upper mixed layer, generating a well mixed water column, especially in regions where tides are strong. In such a regime, I see the usage of isopycnal coordinates quite challenging ... for example, according to Fig. S4 it seems that in shallow areas HYBRID has much less active layers than ZSTAR ...**

➔ We thank the reviewer for the helpful suggestions. We have revised the Methods to clarify how the MOM6 hybrid vertical coordinate works and we added a vertical section with grid interfaces over topography (new Fig. 2).

➔ In each water column we first form a stable, monotonic density profile from temperature, salinity, and pressure and map it to a prescribed set of target densities to obtain isopycnal candidate interface depths. Independently, we define a nominal z-star (z*) grid; in the hybrid this grid is used as a one-sided lower-bound ("floor") constraint. At each level the model takes the deeper of the isopycnal candidate and the z* floor, and then applies the bottom and any optional thickness/depth limits. There is no hard switch between coordinates: the "transition depth" is simply where an isopycnal surface would otherwise lie above the z* floor. Because mixed layers are generally deeper and stratification weaker at higher latitudes, the isopycnal candidates remain shallow for longer and the crossing with the z* floor occurs deeper; hence the transition tends to deepen poleward (consistent with Adcroft et al., 2019).

➔ Regarding shelves, the effective strength of the z* floor is scaled by the local depth in MOM6. Where the bathymetry is shallow, this scaling makes the z* floor very shallow—often only a few meters—so even in winter, if residual stratification exists, interfaces typically follow the target isopycnals over much of the column. Conversely, in fully mixed columns or where the effective z* penetration approaches the local depth, the hybrid reduces to z*.

This also explains the reviewer's observation about "active layers": on shallow shelves many target isopycnals collapse to nearly the same depth and fall below the remapping thickness tolerance, so HYBRID may count fewer active layers than a pure ZSTAR grid, which retains its prescribed z* layers regardless of stratification. We now state this explicitly and clarify in the caption that "active layers" are layers whose thickness exceeds the remapping tolerance.

➔ **(Revised text and Figure)** In the HYBRID configuration, ZSTAR was used to effectively resolve the mixed layer in unstratified regions, providing high resolution where vertical mixing and surface interactions were most significant. Below the mixed layer, isopycnal coordinates were employed to minimize spurious diapycnal mixing and accurately represent the stratified conditions found in deeper waters. The HYBRID in MOM6 is implemented through a column-wise algorithm that combines the strengths of both approaches. In each water column, a stable, monotonic density profile is first derived from temperature, salinity, and pressure and mapped onto a prescribed set of target densities to obtain isopycnal candidate interface depths. Independently, a nominal ZSTAR grid is defined and used as a one-sided lower-bound constraint for each layer. At every vertical level, the model selects the deeper of the two, either the isopycnal candidate or the ZSTAR floor, and then applies bottom and optional thickness/depth limits. As a result, there is no discrete switch between coordinate systems: the transition depth naturally occurs where an isopycnal surface would otherwise lie above the ZSTAR floor. Because mixed layers are deeper and stratification weaker at higher latitudes, the crossing with the ZSTAR floor occurs at greater depths, leading to a poleward deepening of the transition layer. Over continental shelves, the strength of the ZSTAR constraint scales with the local depth. In shallow regions this scaling makes the ZSTAR floor very shallow, so when residual stratification exists, the interfaces tend to follow the target isopycnals through most of the water column. The overall structure of the model interfaces and their interaction with topography are illustrated in Figure 2, which schematically shows how the HYBRID coordinate transitions from ZSTAR near the surface to isopycnal layers in the ocean interior. (Fig.2 and line 169-184)

[Figure]

**Figure 2** Schematic representation of the simulated HYBRID model interfaces and potential density (referenced to 2000 dbar). (a) Meridional section along 148°E showing vertical grid interfaces overlaid on potential density (kg m⁻³). (b) Zonal section along 36°N across the Yellow Sea, illustrating the vertical grid structure adapted to shallow topography.

**L148-157: what formulation of bottom friction is used by the two configurations? What type of lateral boundary conditions are used in the case of the ZSTAR model? How is represented the bottom topography with ZSTAR, with full or partial steps?**

➔ We thank the reviewer for pointing out these missing details. In both ZSTAR and HYBRID configurations, bottom friction is represented using a quadratic drag formulation with a coefficient of Cd = 0.003. For the lateral solid boundaries, free-slip boundary conditions are applied (NOSLIP = False), allowing tangential flow along the walls while preventing normal flow. In the ZSTAR configuration, bottom topography is represented using partial-step layers, ensuring a more realistic bathymetric representation and smoother pressure gradients over steep slopes.

➔ **(Revised text)** The bottom friction was represented using a quadratic drag formulation with a coefficient $C_D$ = 0.003. Lateral solid boundaries employed free-slip conditions, allowing tangential flow along the wall while preventing normal flow. Table 1 provides a summary of the model configurations and parameters. (line 202-204)

**L251: "where u' and v' represent deviations of the zonal and meridional velocity components from their respective means." -> averages over which period?**

➔ We thank the reviewer for this question. The averages were computed over the evaluation period of 2003–2012. We have clarified this point in the revised manuscript.

➔ **(Revised text)** where u' and v^' represent deviations of the zonal and meridional velocity components from their respective means over the evaluation period (2003–2012). (line 280)

**L322-324: It seems to me that in the South China Sea and across the open ocean it is actually the contrary - i.e., HYBRID has a slightly larger positive bias than ZSTAR - see Fig 4 – please clarify.**

➔ We thank the reviewer for the careful reading. We agree that, as shown in Fig. 4, HYBRID tends to have slightly larger positive biases than ZSTAR in the South China Sea and across the open ocean, whereas ZSTAR exhibits larger positive biases in the Yellow Sea. We have revised the sentence to correct this inconsistency.

➔ **(Revised text)** ZSTAR exhibited larger biases than HYBRID in the Yellow Sea, and HYBRID exhibited larger biases than ZSTAR in the South China Sea, the Sea of Okhotsk, and the Kuroshio-Oyashio transition zone. (line 349-351)

**L336: ", with ZSTAR showing stronger positive biases in the open ocean and ..." -> It seems to me that in the open ocean the two model are in the same ballpark, with ZSTAR sightly better than HYBRID in terms of warm bias ... please clarify.**

➔ We thank the reviewer for pointing this out. We agree that in the open ocean the two models perform similarly, with ZSTAR showing slightly smaller positive biases than HYBRID. We have corrected the sentence accordingly in the revised manuscript.

➔ **(Revised text)** The negative bias in the Yellow Sea intensified, exceeding - 1.0 psu, while regional bias patterns persisted, with HYBRID showing stronger positive biases in the open ocean and HYBRID exhibiting more negative biases in the Kuroshio-Oyashio transition zone and Sea of Okhotsk. (line 366)

**L341-344: you already said this at L336-337 - you may want to merge and simplify the text here.**

➔ We thank the reviewer for this suggestion. We agree that the statements at L336–337 and L341–344 were redundant. We have merged them into a single simplified description in the revised manuscript.

**L356: "15 m compared to ZSTAR in the open ocean" -> perhaps "Southern open ocean"?**

➔ We thank the reviewer for the suggestion. We agree that "Southern open ocean" is a more precise description, and we have revised the sentence accordingly in the manuscript.

➔ **(Revised text)** HYBRID exhibited a larger negative bias of about 15 m compared to ZSTAR in the southern open ocean. (line 381)

**L362: Perhaps you were meaning Fig. S2?**

➔ We thank the reviewer for noticing this mistake. Yes, the correct reference is to Fig. S2, and we have corrected it in the revised manuscript.

**Fig. 9 -> Vectors are not clear (I can not see them), perhaps you may consider avoiding plotting them ...**

➔ We thank the reviewer for this helpful suggestion. We agree that the vectors in Fig. 9 were not clearly visible. In the revised manuscript, we have removed the vectors and retained only the shading to improve clarity. (Fig.10)

➔ (Revised Figure)

[Figure]

**Figure 10** Mean surface current speed from GLORYS12, HYBRID, and ZSTAR simulations. (a–c) Spatial distributions of surface current speed with corresponding means and STD. (d) Differences between HYBRID and ZSTAR. (e, f) Biases relative to GLORYS12, including Bias, RMSE, MedAE, and Corr. Contour lines in (d–f) indicate surface current speed biases ranging from -1.0 to 1.0 cm/s at 1.0 cm/s intervals.

**L456: "HYBRID achieved a slightly lower RMSE (138.46 cm²/s²) than ZSTAR (138.86 cm²/s²)" -> to me, they are very similar ...**

➔ We thank the reviewer for this comment. We agree that the RMSE values of HYBRID and ZSTAR are nearly identical, and therefore the statement does not add significant value. We have removed this sentence from the revised manuscript.

**L478: "The Yellow Sea is characterized by the Yellow Sea Bottom Cold Water Mass (YBCWM), a cold and dense water mass that forms near the bottom. However, both configurations showed positive temperature biases exceeding 2°C near the bottom, suggesting limitations in accurately representing YBCWM" -> is this an indication that actually in the Yellow Sea the two models are using a similar vertical discretization – mainly z* levels? Fig. S4 seems to support this conclusion (the difference in active layers in the Yellow Sea is <= 6). This links to the previous comment: it would be great if you could clarify how the hybrid vertical coordinate works in shallow areas (a plot would be very helpful).**

➔ We thank the reviewer for this insightful comment. In MOM6, the HYBRID vertical coordinate system determines the layer interfaces by selecting the deeper of two surfaces—the isopycnal candidate or the ZSTAR floor—at each grid point. This approach allows a continuous transition from z*-like layers near the surface to isopycnal layers in the stratified interior, combining the strengths of both coordinate systems. To examine its behavior in shallow regions such as the Yellow Sea, we compared the model interfaces between HYBRID and ZSTAR during both winter and summer. The results indicate that in winter, isopycnal layers appear mainly in the deeper portion of the water column, whereas in summer they emerge in stratified regions associated with internal wave structures. Thus, the difficulty in reproducing the Yellow Sea Bottom Cold Water Mass (YBCWM) is unlikely to be directly attributable to the vertical coordinate system. Instead, recent studies (e.g., Seelanki et al., 2025, GMD preprint) have shown that the shear-driven mixing parameterization in MOM6 (Jackson et al., 2008) is sensitive to the decay scale, and certain configurations can induce excessive vertical mixing. As the MOM6 version used in this study employs a decay scale associated with stronger mixing, we suggest that the inadequate representation of the YBCWM primarily results from excessive vertical mixing rather than from the choice of vertical discretization.

[Figure]

**Figure 3** Zonal sections of model interfaces and density along 36°N in the Yellow Sea for HYBRID and ZSTAR configurations on 8 February (a–b) and 8 August (c–d). The background color contours represent potential density referenced to 2000 dbar. Black lines denote model layer interfaces, and yellow dashed lines indicate target density surfaces used in the HYBRID configuration.

**Fig. 11: Are the discontinuities in the HYBRID profiles seen in, e.g., panel (b) at around 800m or panel (c) around 400m, due to the transition from z\* to isopycnal levels?**

➔ We thank the reviewer for this helpful observation. The small discontinuities visible in the HYBRID profiles (e.g., around 800 m in panel b and 400 m in panel c) are not related to the z\*–isopycnal transition depth. They appeared in the earlier configuration when the number of active layers was relatively small, which led to segmented-looking profiles. After applying the regional target density array, the number of active layers in the thermocline and intermediate depths increased, yielding smoother and more continuous vertical structures. As a result, these discontinuities were largely mitigated in the updated configuration.

**L620-622: "Overall, the HYBRID configuration tended to overestimate transport, particularly through the Tokara and Tsugaru Straits, while underestimating it in the Soya Strait. In contrast, ZSTAR generally underestimated transport, as observed in the Korea/Tsushima Strait and Soya Straits."Could the authors try to explain possible reasons for this? Could it be due to the fact that both configurations use the same formulation and coefficients for the bottom friction, but the ZSTAR configuration has some additional drag from the lateral boundary conditions (assuming ZSTAR uses a noslip LBC, please clarify)?**

➔ We thank the reviewer for this insightful comment. Both configurations used the same quadratic bottom drag formulation and coefficients, and identical free-slip lateral boundary conditions were applied. Therefore, the transport differences are not related to additional drag from lateral or bottom friction effects.

➔ To better understand the source of the discrepancies, we examined the Tsugaru Strait as a representative narrow passage where HYBRID overestimated transport relative to ZSTAR. Fig. 3 and 4 in this letter presents meridional sections of potential density ($\sigma_2$) and along-strait velocity (U), respectively. The HYBRID configuration exhibits steeper isopycnal slopes and stronger stratification between 41.2° N – 41.4° N and 30–100 m depth, which result in a more pronounced along-strait velocity core compared to ZSTAR. These differences indicate that the enhanced transport in HYBRID arises from stronger baroclinic pressure-gradient forcing associated with sharper density gradients, rather than differences in frictional or boundary formulations.

➔ Mechanistically, the HYBRID vertical coordinate system aligns model layers with isopycnal surfaces derived from temperature and salinity. In narrow straits where the bathymetry changes abruptly and water masses on each side have distinct density structures, isopycnal surfaces can steepen significantly. Under such conditions, HYBRID tends to closely follow

these tilted isopycnals, thereby amplifying the local horizontal density gradient ($\partial\rho/\partial x$) and associated pressure-gradient force. Consequently, the along-strait flow becomes stronger, leading to a slight overestimation of transport relative to ZSTAR.

➔ In contrast, ZSTAR maintains a smoother vertical discretization that damps sharp density gradients and reduces the baroclinic pressure-gradient term, which can lead to underestimated transports. Therefore, the differences between HYBRID and ZSTAR primarily stem from how each coordinate system represents stratification and baroclinic structure, not from drag or lateral boundary effects. (line 758-764)

➔ **(Revised text)** Both configurations showed noticeable differences in volume transport through major straits of the Northwest Pacific. HYBRID tended to overestimate transport through the Tokara and Tsugaru Straits, whereas ZSTAR underestimated it in the Korea/Tsushima. Since both used identical bottom drag and free-slip boundary conditions, these differences are unlikely to result from frictional effects. Instead, the stronger stratification and steeper isopycnal slopes represented by HYBRID (Figs. S8 and S9) may enhance the baroclinic pressure-gradient force and lead to larger transports. However, further investigation is needed to clarify the mechanisms through which different vertical coordinate systems influence transport variability in narrow straits. (line 755-764)

[Figure]

**Figure 4** Meridional section of potential density ($\sigma_2$, referenced to 2000 m) across the Tsugaru Strait, averaged over 2012. (a) ZSTAR, (b) HYBRID, and (c) HYBRID–ZSTAR difference.

[Figure]

**Figure 5** Meridional section of along-strait velocity (U) across the Tsugaru Strait, averaged over 2012. (a) ZSTAR, (b) HYBRID, and (c) HYBRID–ZSTAR difference.

**L652-655: This seems to be in agreement with Wise et al. 2022, that shows that better resolving the bottom topography (terrain following and multi-envelope vertical coordinates versus z\*-levels with partial steps) allows for a more accurate representation of the tides. Also, Graham et al. 2018 and Bruciaferri et al. 2022 showed the detrimental impact of the 10m minimum depth approximation on the tidal propagation, especially in shallow areas. This could explain the consistent underestimation of tidal amplitude in both configurations in the Yellow Sea. Please consider discussing here how the two configurations differ in the representation of the terrain and how this could explain the better performance of HYBRID.**

➔ We appreciate this insightful comment. To better understand why tidal performance differs between the two vertical coordinate systems, we conducted additional analyses, and the detailed results will be presented in a separate manuscript. In this study, we focused on evaluating the HYBRID and ZSTAR configurations against regional observations and found that HYBRID consistently achieved better tidal simulations. The improved performance of HYBRID appears to arise from its more realistic representation of stratification, particularly in shallow and estuarine regions such as the Yellow Sea and the Yangtze River estuary. The HYBRID coordinate preserves vertical density structure more effectively by reducing spurious diapycnal mixing, which helps maintain realistic vertical density gradients that are essential for accurate tidal propagation. In contrast, ZSTAR tends to smooth stratification due to stronger cross-isopycnal mixing, leading to weaker vertical density gradients and an underestimation of tidal amplitude. We have briefly mentioned this point in the revised manuscript and will present the detailed analysis in a dedicated paper.

**L659: "computational efficiency"-> can the authors quantify this for this domain?**

➔ We thank the reviewer for this valuable suggestion. In our Northwest Pacific configuration, the computational efficiency gain in MOM6 can be attributed to its improved numerical stability, which allows for longer stable timesteps compared to MOM5. Specifically, the maximum baroclinic timestep increased from 150 s in MOM5 to 300 s in MOM6, and the tracer timestep increased from 300 s to 900 s. These increases substantially reduced the total number of integration steps required for a given simulation period, directly translating into improved computational efficiency at the same spatial resolution. We have added a clarification of this point in the revised manuscript.

➔ **(Revised text)** Compared with the previous MOM5-based regional model described in Jin et al. (2024), MOM6 demonstrated noticeably higher computational efficiency, primarily due to improvements in numerical stability that allow longer stable timesteps. Specifically, the maximum baroclinic timestep increased from 150 s in MOM5 to 300 s in MOM6, and the tracer timestep increased from 300 s in MOM5 to 900 s in MOM6, substantially reducing the total number of integration steps required for a given simulation period. This enhancement in numerical stability directly translates into greater computational efficiency under the same model resolution. To further assess how the choice of vertical coordinate system influences computational cost within MOM6, we compared the HYBRID and ZSTAR configurations using the same supercomputer node and identical processor layouts (42 × 40 PE decomposition, with 536 PEs masked through land processor masking). The ZSTAR configuration required an average of 23 h per simulated year, whereas HYBRID completed the same simulation in 20.2 h, indicating that ZSTAR consumed approximately 2 h more. This difference primarily reflects the number of active vertical layers in each configuration: ZSTAR maintained a consistently high number of layers across most of the domain, while HYBRID adaptively reduced active layers in weakly stratified and shallow regions (Fig. S5), leading to fewer computations (line 661-674)

**L683: "lower salinity biases" -> but the temperature biases seem to me much larger (Fig 12e) – please clarify.**

➔ We thank the reviewer for this valuable comment. As noted, the temperature biases in Fig. 12e were indeed more pronounced when the HYBRID configuration employed the global target density array originally designed for the OM4.0 global model. This global setup tended to produce fewer active layers in the upper ocean, particularly in stratified regions, limiting vertical resolution and leading to excessive warming near the surface. Following

the reviewer's suggestion, we tested a regional target density configuration optimized for the Northwest Pacific domain. The regional target densities increased the number of active layers in the thermocline and upper ocean, thereby improving the vertical representation of temperature and mitigating the positive temperature biases previously observed in the HYBRID.

**L694-695: "While such processes facilitate surface-to-interior interactions, they likely contributed to the observed temperature and salinity biases (Figs. 12 and 13)." -> could the authors please clarify a possible mechanism for this?**

➔ We thank the reviewer for this helpful comment. In the initial experiments using the global target density array from OM4.0, temperature and salinity biases were evident within the Subtropical Mode Water (STMW) region, primarily due to the insufficient number of active layers. These conditions facilitated overly strong surface-to-interior exchange, which we originally described in the cited sentence. After applying the regional target density array optimized for the Northwest Pacific, the number of active layers in the upper thermocline increased substantially, leading to a much improved representation of STMW and a reduction in both temperature and salinity biases. Since this improvement effectively mitigated the previously noted issue, we have removed the corresponding sentence ("While such processes facilitate surface-to-interior interactions, they likely contributed to the observed temperature and salinity biases") from the revised manuscript. We sincerely appreciate the reviewer's suggestion, which guided us to perform this refinement and ultimately improve the model's vertical structure and bias representation.

**L701-702: "This reduction in active layers likely contributed to the increased temperature and salinity biases observed in HYBRID, underscoring the challenges of using isopycnal coordinates in high-latitude environments." -> I think Fig S4 shows that HYBRID has fewer active layers not just at high latitudes, but everywhere .... As I mentioned before, are we sure these are intrinsic difficulties of the hybrid z*-isopycnal vertical coordinate and not due to the not-optimal configuration of the HYBRID vertical grid?**

➔ We thank the reviewer for this thoughtful comment. We agree that the reduced number of active layers shown in Fig. S4 is not an intrinsic limitation of the HYBRID coordinate itself. In our follow-up experiments, the application of a regional target density array increased the number of active layers, especially in the upper and thermocline layers. However, even with this improvement, temperature and salinity biases persisted or became slightly larger

in some regions. This suggests that the remaining biases are not solely due to the number of active layers but may instead reflect regional mismatches between the prescribed target densities and the actual density structure of the Northwest Pacific. In other words, although the vertical resolution was enhanced, the target density distribution still requires further regional tuning to better represent local water masses. It may therefore be necessary to consider applying spatially varying (three-dimensional) target density fields that can better adapt to regional hydrographic conditions and further improve the fidelity of the HYBRID configuration.

**L703-704: "modifications to the target density profile could enhance vertical resolution and better capture stratification" -> could the authors clarify why they chose to use a global setup in a regional model?**

➔ We thank the reviewer for this constructive comment and the opportunity to clarify our initial choice. The global target density profile from OM4.0 (Adcroft et al., 2019) was originally adopted to ensure methodological continuity and comparability with the well-validated global MOM6 configuration. This approach provided a consistent baseline for evaluating the vertical coordinate performance in our regional setup without introducing new sources of uncertainty at the initial stage. However, as the reviewer correctly pointed out, the global configuration was not fully optimized for the hydrographic conditions of the Northwest Pacific. To address this, we subsequently developed and tested a regional target density profile tailored to the local density structure. The regional configuration improved the vertical resolution and stratification representation, particularly in the upper ocean, and helped reduce temperature and salinity biases.

**L725: "HYBRID's improved tidal amplitude simulation is likely linked to its enhanced representation of stratification." -> or to a better representation of flow-topography interactions in comparison to ZSTAR?**

➔ We thank the reviewer for this constructive comment. While improved representation of flow–topography interactions can certainly influence tidal performance, we believe that the primary factor behind the enhanced tidal amplitude in the HYBRID configuration is its more realistic depiction of stratification. The Yellow Sea, in particular, exhibits strong seasonally varying stratification that significantly modulates tidal propagation and dissipation (Kang et al., 2002; Liu et al., 2019). In such regions, accurate representation of vertical density gradients is crucial because stratification controls internal tide generation and modifies barotropic energy conversion and dissipation pathways. The HYBRID coordinate system

better preserves these vertical density structures by reducing spurious diapycnal mixing, thereby improving the vertical phase structure of tides and their surface expression.

**L729-731: "further investigation is needed to clarify the mechanisms through which different vertical coordinates influence tidal dynamics, particularly the generation, propagation, and dissipation of baroclinic tides." -> The 10m minimum depth approximation could be one possible candidate (see Graham et al. 2018 and Bruciaferri et al. 2022).**

➔ We thank the reviewer for this insightful comment and for pointing out the potential influence of the 10 m minimum depth approximation. We agree that this approximation can affect tidal propagation in shallow regions, as shown by Graham et al. (2018) and Bruciaferri et al. (2022). However, in our experiments, the difference in tidal performance between the HYBRID and ZSTAR configurations appears to be primarily related to their contrasting representations of stratification rather than the minimum-depth setting. The HYBRID coordinate system better preserves vertical density gradients and reduces spurious diapycnal mixing, which leads to a more realistic stratification structure that directly affects tidal energy propagation and dissipation. In contrast, the ZSTAR configuration tends to produce smoother and weaker stratification, resulting in reduced tidal amplitudes, particularly in the Yellow Sea.

**Reference**

Kang, S. K., Foreman, M. G., Lie, H. J., Lee, J. H., Cherniawsky, J., & Yum, K. D. (2002). Two-layer tidal modeling of the Yellow and East China Seas with application to seasonal variability of the M2 tide. Journal of Geophysical Research: Oceans, 107(C3), 6-1. doi: 10.1029/2001JC000838

Liu, K., Sun, J., Guo, C., Yang, Y., Yu, W., & Wei, Z. (2019). Seasonal and spatial variations of the M2 internal tide in the Yellow Sea. Journal of Geophysical Research: Oceans, 124(2), 1115-1138. doi: 10.1029/2018JC014819

**Author's General Response to Reviewer Matthew Harrison**

We sincerely thank the reviewer for their thoughtful and constructive comments. The reviewer's questions and suggestions have helped us improve the clarity and depth of our analysis, particularly regarding the influence of the vertical coordinate system on stratification, MLD, and strait transports. We have carefully revised the manuscript to address all comments, and we believe these revisions have strengthened the overall quality of the paper. Detailed responses to each comment are provided below.

We have incorporated these new results into the revised manuscript. Overall, the reviewer's comments have led to a more balanced and comprehensive assessment of the vertical coordinate sensitivity in the regional MOM6 framework.

The detailed responses to the reviewer's specific comments are provided below.

1. **Figure 1: Spelling correction. "East Chania Sea" should be "East China Sea"**

➔ We thank the reviewer for pointing this out. The spelling error in Figure 1 has been corrected — "East Chania Sea" has been revised to "East China Sea."

2. **L110: Reference Ilicak, 2012, Ocean Modeling.**

➔ We thank the reviewer for the suggestion. The reference to Ilicak et al. (2012, Ocean Modelling) has been added in the revised manuscript. (line 109)

➔ (Revised text) Spurious mixing in ocean models is a major concern, as it introduces an unphysical process that unintentionally increases total mixing beyond the prescribed and parameterized levels (Griffies et al., 2000; Ilicak et al., 2012; Gibson et al., 2017). (line 109)

3. **L143: How does limiting the depth to 5000m reduce the model's computational burden?**

➔ We thank the reviewer for this insightful comment. In the original manuscript, the term computational efficiency was used to describe the benefit of limiting the maximum model depth to 5000 m. However, as the reviewer correctly points out, this change does not primarily reduce computational cost but rather improves the efficiency of the vertical grid design. Specifically, limiting the depth avoids allocating unnecessary vertical layers in dynamically inactive deep basins and allows more effective use of vertical resolution in the

upper and intermediate layers. Accordingly, we have replaced computational efficiency with efficiency of vertical grid utilization in the revised manuscript to better reflect the intended meaning.

➔ (**Revised text)** The minimum bathymetric depth was set to 10 m to account for tidal variations, as wetting and drying were not employed, while the maximum depth was limited to 5,000 m to enhance efficiency of vertical grid utilization. (line 154)

4. **The "background" vertical viscosities and diffusivities are less than molecular (1.e-6 m2 s-1) values. Are MOM6 configurations typically using such small values? This appears to be a departure from typical ocean configurations which would typically rely on background values an order of magnitude higher. Please discuss the rationale for using such a small value.**

➔ We thank the reviewer for this valuable comment. The small background vertical viscosity and diffusivity values ($1 \times 10^{-6}$ m² s⁻¹) used in this study follow the rationale that in MOM6, these parameters represent only a residual background contribution, as the dominant mixing processes are already parameterized through the energetic Planetary Boundary Layer (ePBL; Reichl & Hallberg, 2018) and the shear-driven mixing scheme (Jackson et al., 2008). Because these parameterizations explicitly represent vertical mixing and turbulence, the background values mainly serve to maintain numerical stability rather than to control mixing intensity. Sensitivity tests comparing $1 \times 10^{-5}$ m² s⁻¹ and $1 \times 10^{-6}$ m² s⁻¹ showed no significant differences in the large-scale circulation or tracer distribution, and therefore the smaller value was adopted for consistency and to minimize artificial diffusion.

5. **Table 1 could be reformatted, since the third column is empty with the exception of the first row.**

➔ We thank the reviewer for this suggestion. Table 1 has been reformatted to remove the empty third column and improve readability. The revised version now presents the information in a more concise and visually clear layout.

6. **L201: For clarification, are you applying discharge adjustments at the Yangtze river only?**

➔ We thank the reviewer for this helpful question. In the initial setup, discharge adjustments

were applied only to the Yangtze River, which has the largest freshwater influence in the Northwest Pacific region. However, despite this correction, the model still exhibited low-salinity biases along the Chinese coast. To address this issue, we recently extended the discharge bias correction to include the Yellow River as well.

7. **L237-241: This paragraph is redundant with the previous one.**

➔ We thank the reviewer for pointing this out. The paragraph at Lines 237–241 overlapped with the preceding one, and we have removed the redundancy by merging the relevant information into a single, concise paragraph in the revised manuscript.

8. **L352-366: The differences in wintertime MLD between HYBRID and ZSTAR are most prominent South of the Kuroshio extension between 25N-35N and in the Okhotsk Sea. In the first case, is this a result of enhanced stratification below the actively mixed layer prior to the onset of wintertime convection, for example? In the latter case, is this a result of poor representation (fewer active layers) due to the use of sigma2? It would be helpful to see the seasonal evolution of the stratification and/or the actual internal layer representation in the model for these regions. These questions could be addressed later in the text as well.**

➔ We thank the reviewer for this insightful question. In the region south of the Kuroshio Extension (25°–35°N), the comparison of buoyancy frequency squared ($N^2$) between ZSTAR and HYBRID (Fig. S6) indicates that both configurations capture strong stratification beneath the mixed layer, but the vertical structure differs. HYBRID shows stronger stratification slightly deeper (below about 100 m) from late summer to early winter, whereas ZSTAR exhibits stronger stratification just below the mixed layer (around 50–100 m). The deeper stratification maximum in HYBRID stabilizes the upper ocean and suppresses excessive convective deepening during winter, resulting in a shallower and more realistic MLD compared to ZSTAR, which tends to overestimate MLDs because of weaker near-surface stratification. Although the magnitude of the $N^2$ differences ($\sim 10^{-5}$ $s^{-2}$) is modest, it is seasonally consistent and sufficient to influence the mixed-layer evolution.

[Figure]

**Figure 6** Seasonal evolution of buoyancy frequency squared (N², s⁻²) averaged over 25°–35°N and 140°–160°E for (a) HYBRID, (b) ZSTAR, and (c) their difference (HYBRID − ZSTAR).

➔ In the Sea of Okhotsk, the MLD differences primarily arise from the vertical layer representation (Fig. S7). In this region, the σ₂-based HYBRID coordinate produces thicker layers in weakly stratified waters, reducing vertical resolution below roughly 80–200 m and leading to slightly deeper MLDs than ZSTAR. (line 698-708)

[Figure]

**Figure 7** Model interfaces along 152°E in the Sea of Okhotsk for (a) HYBRID and (b) ZSTAR.

➔ **(Revised text)** In the Kuroshio region (25°–35°N), both configurations captured strong stratification beneath the mixed layer, but the vertical structure differed. HYBRID exhibited stronger stratification slightly deeper (below ~100 m) from late summer to early winter, while ZSTAR showed stronger stratification just below the mixed layer (around 50–100 m) (Fig. S6). The deeper stratification maximum in HYBRID stabilized the upper ocean and limited wintertime convective deepening, resulting in a shallower and more realistic MLD

compared to ZSTAR, which tended to overestimate MLDs due to weaker near-surface stratification. In the Sea of Okhotsk, MLD differences mainly arose from the vertical layer representation: the σ₂-based HYBRID coordinate formed thicker layers in weakly stratified waters (Fig. S7), reducing vertical resolution below approximately 80–200 m and leading to slightly deeper MLDs than ZSTAR. These results suggest that the way each vertical coordinate system represents stratification strength and layer spacing substantially influences the simulated MLD structure across the Northwest Pacific. (line 698-708)

9. **Fig15: salinity differences on sigma2 can be mostly estimated from the temperature differences, so this figure and the previous one are redundant. Suggest removing this figure.**

➔ We thank the reviewer for this helpful suggestion. Since the salinity differences on σ₂ surfaces were largely consistent with the temperature-induced density variations, we agree that Fig. 15 conveyed redundant information. To streamline the main text while retaining the supporting analysis, this figure has been moved to the Supplementary Material, and only a brief reference to it remains in the revised manuscript.

10. **L592. Did the authors evaluate simulations without explicit tides and using parameterized tides instead and did this reveal differences in YBCMW?**

➔ We thank the reviewer for this insightful question. In our previous MOM5-based configuration, explicit tidal forcing was not included, and tidal effects were represented through parameterized mixing schemes. To examine their impact on the formation of the Yellow Sea Bottom Cold Water Mass (YBCWM), additional experiments were conducted. The results showed that, without data assimilation, the model using parameterized tides was unable to reproduce the YBCWM structure. This indicates that the explicit representation of tides and associated mixing processes plays an essential role in maintaining the cold, dense bottom water in the Yellow Sea.

11. **L622. Can you provide additional comments on the role of the vertical coordinate in impacting the volume transport through straits? The connection is not immediately obvious.**

➔ We thank the reviewer for this valuable question. As also noted by another reviewer, the

vertical coordinate system can influence the simulated volume transport through narrow straits by modifying the local representation of stratification and pressure gradients. In our simulations, both configurations used identical bottom drag formulations and free-slip lateral boundary conditions; therefore, the transport differences are not attributed to frictional effects. Instead, sectional analyses across the Tsugaru Strait revealed that HYBRID preserved steeper isopycnal slopes and stronger density gradients than ZSTAR, which may enhance the baroclinic pressure-gradient force and lead to stronger along-strait velocities. In narrow passages where bathymetry changes abruptly and density structures differ across the strait, HYBRID's isopycnal alignment can locally amplify the horizontal density gradient ($\partial\rho/\partial x$) and thus the transport. Conversely, ZSTAR's smoother vertical discretization dampens sharp density gradients, producing weaker baroclinic pressure gradients and smaller transports. These findings suggest that the differences in volume transport between HYBRID and ZSTAR primarily arise from how each vertical coordinate system represents stratification and baroclinic structure, rather than from differences in friction or boundary formulations. We have clarified this interpretation and added the relevant explanation in the revised manuscript.

➔ **(Revised text)** Both configurations showed noticeable differences in volume transport through major straits of the Northwest Pacific. HYBRID tended to overestimate transport through the Tokara and Tsugaru Straits, whereas ZSTAR underestimated it in the Korea/Tsushima. Since both used identical bottom drag and free-slip boundary conditions, these differences are unlikely to result from frictional effects. Instead, the stronger stratification and steeper isopycnal slopes represented by HYBRID (Figs. S8 and S9) may enhance the baroclinic pressure-gradient force and lead to larger transports. However, further investigation is needed to clarify the mechanisms through which different vertical coordinate systems influence transport variability in narrow straits. (line 758-764)

[Figure]

**Figure 8** Meridional section of potential density (σ2, referenced to 2000 m) across the Tsugaru Strait, averaged over 2012. (a) ZSTAR, (b) HYBRID, and (c) HYBRID–ZSTAR difference.

[Figure]

**Figure 9** Meridional section of along-strait velocity (U) across the Tsugaru Strait, averaged over 2012. (a) ZSTAR, (b) HYBRID, and (c) HYBRID–ZSTAR difference.

**Author's General Response to Reviewer #3**

We sincerely thank the reviewer for their careful reading and constructive feedback. The reviewer's comments were very helpful in improving the clarity, completeness, and presentation quality of the manuscript. We have added appropriate references and examples where needed, refined figure readability, and clarified several descriptions to better explain the physical interpretation of our results. In addition, we have added a new part discussing the computational cost differences between the HYBRID and ZSTAR configurations, as suggested (line 661-674). These revisions have enhanced both the accuracy and readability of the paper. Detailed responses to each comment are provided below.

**Line 200-204: My interpretation is that the bias correction was applied only to the Yangtze River. If so, why was this only done for this river? The authors recognize in the results and conclusions the biases in salinity, I wonder why the authors did not apply the same correction to other rivers to potentially help with the SSS biases shown.**

➔ We thank the reviewer for this question. In the initial configuration, discharge bias correction was applied only to the Yangtze River, which exerts the strongest freshwater influence in the Northwest Pacific region. However, despite this correction, the model still exhibited low-salinity biases along the Chinese coast. To address this issue, we have recently extended the bias correction to include the Yellow River as well. We agree that applying discharge adjustments to multiple river sources can further reduce coastal salinity biases, and we plan to incorporate additional river discharge datasets as they become available.

**Line 270: "The K-KE regions..." as obvious as it may seem, you have not defined what this stands for.**

➔ We thank the reviewer for pointing this out. The term "K-KE" has been defined in the revised manuscript as referring to the Kuroshio and its Extension regions.

➔ **(Revised text)** The Kuroshio and its Extension regions were defined not only based on areas where the climatological surface current speed (line 299)

**Lines 249-254: It would be useful to define the period over which the average of the u and v velocities is calculated.**

➔ We thank the reviewer for this helpful comment. The averaging period for the zonal (u) and meridional (v) velocity components has been clarified in the revised manuscript as 2003–2012, consistent with the main evaluation period used throughout the analysis.

➔ (Revised text) where u' and v^' represent deviations of the zonal and meridional velocity components from their respective means over the evaluation period (2003–2012). (line 280)

**Figures 2, 3 and others: I found it hard to distinguish land and contours on areas with high contour line density in the panels showing differences between products and model output because the color used for land and contours is the same. I would encourage changing land color to black or setting the contour line color to a color other than gray.**

➔ We thank the reviewer for this helpful suggestion. We agree that the use of similar colors for land and contour lines reduced visual clarity in the difference plots. In the revised figures (Figs. 2, 3, and others), we have improved readability by changing the land color to dimgray and slightly adjusting the contour line contrast to ensure clearer distinction between land and contours.

**Section 3.4: It would be useful to include here and/or in Section 4 a discussion on how the volume transport differences come about with the different vertical grids. It seems that, while not perfect, z-star was consistently better than the hybrid grid.**

➔ We thank the reviewer for this valuable suggestion. This comment aligns with another reviewer's question regarding how the vertical coordinate system influences the simulated volume transport through major straits. We have clarified this connection in the revised manuscript (Section 4.2), explaining that both configurations used identical bottom drag formulations and free-slip lateral boundary conditions; thus, the transport differences mainly arise from how each vertical coordinate represents stratification and the associated baroclinic pressure gradients. In particular, HYBRID tends to preserve steeper isopycnal slopes and stronger density gradients, which may locally enhance the pressure-gradient force and result in stronger along-strait velocities, while ZSTAR's smoother vertical discretization dampens these gradients and yields weaker transports.

➔ **(Revised text)** Both configurations showed noticeable differences in volume transport through major straits of the Northwest Pacific. HYBRID tended to overestimate transport through the Tokara and Tsugaru Straits, whereas ZSTAR underestimated it in the Korea/Tsushima. Since both used identical bottom drag and free-slip boundary conditions,

these differences are unlikely to result from frictional effects. Instead, the stronger stratification and steeper isopycnal slopes represented by HYBRID (Figs. S8 and S9) may enhance the baroclinic pressure-gradient force and lead to larger transports. However, further investigation is needed to clarify the mechanisms through which different vertical coordinate systems influence transport variability in narrow straits. (line 755-764)

[Figure]

**Figure 1** Meridional section of potential density ($\sigma_2$, referenced to 2000 m) across the Tsugaru Strait, averaged over 2012. (a) ZSTAR, (b) HYBRID, and (c) HYBRID–ZSTAR difference.

[Figure]

**Figure 2** Meridional section of along-strait velocity (U) across the Tsugaru Strait, averaged over 2012. (a) ZSTAR, (b) HYBRID, and (c) HYBRID–ZSTAR difference.

**Lines 658-660: provide reference and/or examples.**

➔ We thank the reviewer for this helpful comment. Relevant references and supporting examples have been added in the revised manuscript to substantiate this statement.

**(Revised text)** MOM6 introduced significant improvements in computational efficiency, numerical stability, and flexibility in vertical coordinate selection, enabling a more advanced representation of oceanic processes (Jackson et al., 2008; Reichl and Hallberg, 2018; Reichl and Li, 2019; Adcroft et al., 2019; Griffies et al., 2020)

---

## Author Response (AR2)

**Reviewer 1**

We sincerely thank the reviewer for the positive assessment of our revised manuscript and for the helpful minor suggestions. We have carefully addressed the two comments as follows:

**1) Abstract and Section 2.1**
We appreciate the reviewer's suggestion.

- We have **updated the abstract** to briefly mention the limitations of both configurations in reproducing winter SST.
- In **Section 2.1**, we have restored the description indicating that the **surface layer thickness is 2 m**, which was unintentionally removed during revision.

**2) Clarification of the target density specification**
We thank the reviewer for pointing out this inconsistency. We have revised the text to clearly state that:

- The **target density used in this study ranges from 1010 to 1037.2479 kg/m³ referenced at 2000 dbar**,
- The previous sentence referencing the SIGMA2 10.00–38.00 kg/m³ range has been removed to avoid confusion.

We appreciate the reviewer's thorough reading and constructive feedback, which helped further improve the clarity of the manuscript.

**Reviewer 2**

We sincerely thank the reviewer for the positive assessment of our revised manuscript and for recommending it for publication. We also appreciate the reviewer's constructive suggestion regarding the figure presentation. We have addressed the comment as follows:

**1) Revision of difference panels in Figures 3–11**
We agree with the reviewer that the high density of contour lines made it difficult to distinguish land from the closely spaced contours in the difference panels.

- To improve visual clarity, we have removed the gray and black contour lines entirely from all difference plots.

We appreciate the reviewer's helpful feedback, which improved the readability and overall quality of the figures.